# Understanding the role of Shroom3 in the developing mouse myocardium

Jennifer L. Carleton [1,2]*, Rami R. Halabi[1,2], Jessica A. Willson[1,3], Timothy F. Plageman Jr.[4], Darren Bridgewater[5], Qingping Feng [1,2,6], Thomas A. Drysdale[1,2,3]

1 Children's Health Research Institute, Victoria Research Labs, London, Ontario, Canada, 2 Department of Physiology & Pharmacology, The University of Western Ontario, London, Ontario, Canada, 3 Department of Paediatrics, The University of Western Ontario, London, Ontario, Canada, 4 College of Optometry, The Ohio State University, Columbus, Ohio, United States of America, 5 Department of Pathology & Molecular Medicine, Faculty of Health Sciences, McMaster University, Hamilton, Ontario, Canada, 6 Department of Medicine, The University of Western Ontario, London, Ontario, Canada

☯ These authors contributed equally to this work.
* jcarlet@uwo.ca

## Abstract

Loss of actin cytoskeleton control can hinder integral developmental and physiological processes and can be the basis for a subset of developmental defects. SHROOM3 is an actin binding protein, best characterized as being essential for neural tube closure in vertebrates. *Shroom3* expression has also been identified in the developing heart, with some associated congenital heart defects. Here we show that the expression pattern of *Shroom3* in the developing and adult mouse heart is specific to the myocardium. Using a gene trap line, we show that embryos with homozygous full-body *Shroom3* loss die at birth due to exencephaly but also show congenital heart defects. This includes ventricular septal defects, semilunar valve abnormalities, and ventricle wall thinning. Adult mice heterozygous for *Shroom3* loss also show ventricular thinning due to decreased cardiomyocyte size. To explore if SHROOM3 is operating in a cell autonomous manner in the cardiomyocytes, we utilized a floxed *Shroom3* mouse line, allowing for spatial and temporal control of *Shroom3* loss. Using an Nkx2–5-Cre recombinase, we targeted *Shroom3* loss to the myocardium of the developing heart. Neonate pups with myocardial specific *Shroom3* loss showed no significant impact on heart development, including no septal or valve defects, no ventricular thinning, and no change in viability into adulthood. Adult mice with myocardial specific *Shroom3* loss showed no ventricular thinning and no change in cardiomyocyte size. These results show that the heart defects seen in full-body *Shroom3* loss do not arise from myocardial specific loss. Rather, other cell types expressing *Shroom3*, such as the cardiac neural crest cells, may be directly contributing to cardiac development.

**Data availability statement:** All relevant data are within the paper and its Supporting information files.

**Funding:** Natural Sciences and Engineering Research Council of Canada Award Number: RGPIN-2016-06536 Canadian Institutes of Health Research Award Number: MOP133593 The funders had no role in study design, data collection and analysis, decision to publish, or preparation of the manuscript.

**Competing interests:** The authors have declared that no competing interests exist.

## Introduction

Congenital heart defects (CHDs) are seen in approximately 1% of human births, making them one of the most prevalent congenital disorders in newborns [1–3]. With the progression of specialized surgical and diagnostic techniques, over 90% of children born with CHDs survive to adulthood [4,5]. The ability to reach adulthood presents a new challenge to researchers and clinicians: how does this impact our understanding of the potential heritability of CHDs? As a candidate gene for CHDs, SHROOM3 is necessary for proper heart development. Full-body loss of *Shroom3* in the developing mouse shows a spectrum of CHDs including ventricular septal defects, double outlet right ventricle, and thinning of the myocardium in the left ventricle [6]. *Shroom3* missense mutations have also been identified and are associated with heterotaxy and hemophagocytic lymphohistiocytosis in patient populations [7,8].

The SHROOM3 protein contains two major functional domains: the central Apx/Shrm-specific domain 1 (ASD1) allows for direct binding with f-actin, and the C-terminal ASD2 domain allows for direct binding with ROCK [9–16]. These functions allow for myosin light chain phosphorylation, inducing actomyosin constriction and thus SHROOM3-induced apical constriction [10,17–23]. There have also been indications that SHROOM3 is able to drive the re-distribution of γ-tubulin, allowing for apical-basal elongation, however the basis for this activity is less understood [20,24]. This cell shape change is described in polarized epithelial cells, where it plays a pivotal role in many developmental processes, most notably neural tube folding and closure, but also lens pit invagination, directional gut tube looping, and maintenance of glomerular structural integrity in the kidney [9–11,14,16,18,22,23,25–28]. In humans, *Shroom3* loss-of-function variants have been linked with anencephaly, spina bifida, and cleft lip and palate [29,30]. While promoting rolling of the neural plate into the neural tube is the best-established function of SHROOM3, epithelial folding has not been documented during mammalian heart development. As such, the functional contributions of *Shroom3* to cardiac morphogenesis remain unclear. We set out to understand where and when *Shroom3* is expressed in the mouse heart, and what the consequences of *Shroom3* loss are on cardiac development.

In this study we document *Shroom3* expression specifically in the myocardium of the developing and adult mouse heart. We found that full-body loss of *Shroom3* produces CHDs in embryos, similar to what has been described in Durbin *et al.* 2020. Adult mice with full-body *Shroom3* loss show ventricular thinning due to decreased cardiomyocyte size. Here we also utilize a novel mouse line containing a floxed *Shroom3* allele, allowing for spatial and temporal control of *Shroom3* loss. In combination with a cardiac specific *Nkx2–5* promoter driven Cre recombinase, we have selectively eliminated *Shroom3* in the myocardium of the developing heart. Despite the loss, there were no notable CHDs or long-term effects on adult hearts seen in these mice. These results indicate that while SHROOM3 plays a role in proper heart development, myocardial SHROOM3 is not necessary for this role.

## Methods

### Mouse lines

All animal experiments were approved by Western University ACC (Protocol #015–2011 and #064–2019). All procedures were approved by the Council on Animal Care at The University of Western Ontario, in accordance with the guidelines of the Canadian Council on Animal Care.

Mouse line B6.129S4-Shroom3[Gt(ROSA53)Sor/J] was purchased from The Jackson Laboratory on a C57BL⁄6 background. This line was generated using a gene trap assay to insert a SaβgalCrepA cassette under the control of the endogenous *Shroom3* promoter [31]. This line will be referred to as *Shroom3*[Gt]. A schematic for the gene trap vector insertion is depicted in S1A Fig.

Mouse line Tg(Nkx2–5-cre)9Eno (MGI:3514028) was donated to us by the laboratory of Dr. Qingping Feng and was maintained on a C57BL⁄6 background. This transgenic line shows the highest level of expression in the myocardium of the developing heart, beginning in the developing heart tube. Later during development this expression is localized to the left and right ventricular myocardium [32]. This transgenic line will be referred to as *Nkx2–5-Cre.*

Cyagen Biosciences was contracted to create a novel floxed *Shroom3* allele mouse line. Using C57BL⁄6 embryonic stem cells, two loxP sites were inserted into the *Shroom3* allele using a targeting vector containing a Neo cassette flanked by self-deletion anchor sites. These loxP sites were inserted to flank exon 5 of the *Shroom3* gene on mouse chromosome 5. Upon Cre-mediated recombination of the floxed allele, exon 5 is excised, resulting in a constitutive knockout (KO) allele and a loss of function of the *Shroom3* gene. This floxed *Shroom3* mouse line has been named C57BL/6-Shroom3[tm1Shrc]. A schematic for the targeting vector, targeted allele, and KO allele is depicted in S1B Fig. This line will be referred to as *Shroom3*[fl]. A more in-depth description of the targeting vector and line generation can be found in Herstine *et al.*, 2025.

Pregnant dams were assessed for a vaginal plug in the morning following mating. The morning that the vaginal plug was seen was designated as embryonic day 0.5 (E0.5). For embryonic studies, pregnant dams were euthanized using $CO_2$ gas inhalation and embryos were dissected at the required developmental time-point. For neonate studies, pups were taken on the day of birth and euthanized via decapitation. For adult studies, mice were euthanized using $CO_2$ gas inhalation at the desired age according to standard ACC guidelines.

### Genotyping

Adult, neonate, and embryonic mouse genotypes were determined using tail tip DNA. Tissue specific genotyping was carried out using small pieces of the tissues of interest. DNA was isolated with phenol/chloroform extraction and gradient ethanol washed. *Shroom3*[Gt] LacZ cassette insertion primer pairs: GT-F: 5'-GGTGACTGAGGAGTAGAGTCC-3' and GT-R: 5'-GAGTTTGTGCTCAACCGCGAGC-3'. *Nkx2–5-Cre* transgene primer pairs: NCre-F: 5'-ACTGATTTCGACCAGGTT CGTT-3' and NCre-R: 5'-CCCAGGCTAAGTGCCTTCTCTA-3'. *Shroom3*[fl] loxP insertion primer pairs: loxP-F: 5'-CCAGGAAGGTTGCCAGAGTCTAGCT-3' and loxP-R: 5'-CTGTCCGTTGTGGATGCTCGTG-3'. All PCR products were run on a 1% agarose gel with 1 Kb Plus DNA Ladder (Invitrogen, 10787018).

### mRNA isolation, cDNA synthesis, RT-PCR

mRNA isolation from tissues of interest was performed with Trizol/chloroform extraction. 1ug of mRNA was used for cDNA synthesis with the qScript cDNA synthesis kit (QuantaBio, 95047). TBP was used as the housekeeping gene for RT-PCR. *Shroom3*[fl] loxP recombination primer pairs, to be used with cDNA: ExonRec-F: 5'-TATCTCAGGGCACA ATGGGC-3', ExonRec-R: 5'-GGAGAAAGGAGATGGCAGGG-3', and ExonRec-Ko: 5'-TTCCTGCTGAGAGTGGCCTA-3'. TBP housekeeping gene primer pairs, to be used with cDNA: TBP-F: 5'-ACAGGAGCCAAGAGTGAAGA-3' and TBP-R: 5'-CTACTGAACTGCTGGTGGGT-3'.

## Wholemount X-gal staining and clearing

Whole embryos from E9.5 to E13.5, or excised hearts of E14.5 embryos and older were used for wholemount X-gal staining. Heterozygous and wild type control samples were stained in parallel. n = 10 for embryonic and adult stages. A mixed cohort was used for all X-gal staining.

Tissues were fixed for 20 minutes to 2 hours depending on specimen thickness in 4% PFA, and washed overnight at 4°C in PBST. Samples were stained with X-gal staining solution (0.5 mg/ml X-Gal in DMF, 400 µM potassium ferricyanide, 400 µM potassium ferrocyanide, 2 mM MgCl2) in PBST, at 37°C overnight or until the desired level of staining was developed. Tissues were post fixed in 4% PFA for 2 hours. Samples were then cleared as described in [33] with the following modifications: tissues were cleared in scintillation vials for one day each in 20% and 50% glycerol (v/v) with 1% KOH (w/v) in PBS at room temperature, followed by 2–3 days in 80% glycerol (v/v) with 1% KOH (w/v) in PBS at 37°C and finally at 100% glycerol (v/v) with 1% KOH (w/v) in PBS at room temperature until desired translucencies were achieved.

## Hematoxylin and eosin staining

Embryonic, neonate, and adult hearts were fixed in 4% PFA and embedded in paraffin. Tissues were sectioned at 5 µm and stained with CAT hematoxylin (BioCare Medical), followed with Tasha's Bluing Solution (BioCare Medical), and then Eosin Y (0.25% in ethanol, Fisher Scientific). A mixed cohort was used for all H&E staining.

## Congenital and postnatal heart morphology assessment

Adult, neonate, and embryonic hearts were collected, processed, and H&E stained as described above. *Shroom3*<sup>Gt</sup> hearts were serially sectioned in the transverse plane. *Nkx2–5-Cre;Shroom3*<sup>fl</sup> hearts were serially sectioned in the frontal plane. A mixed cohort was used for all morphological analysis.

Ventricular septal defects (VSDs) were counted when a break in the ventricular septum was found in the histological section. Membranous VSDs were classified as those found within the top third of the cardiac septum. Muscular VSDs were counted when the defect was localized to the thicker, muscular portion of the ventricular septum. n = 24 for *Shroom3*<sup>+/+</sup> and *Shroom*<sup>+/Gt</sup> E18.5 embryos and 33 for *Shroom3*<sup>Gt/Gt</sup> E18.5 embryos. n = 23 for *Nkx2–5-Cre;Shroom3*<sup>fl/fl</sup> and *Shroom3*<sup>fl/fl</sup> neonate mice and 8 for *Nkx2–5-Cre;Shroom3*<sup>+/fl</sup> neonate mice.

Semilunar valve defects were based on visual observation. Thick pulmonary valves and malformed aortic valves were comparable to the abnormalities seen in the ADAM17 knockout mouse which exhibits semilunar valve defects [34]. This was used as a guide for valve defects, in addition to valve comparison in wild type littermate controls. n = 18 for *Shroom3*<sup>+/+</sup> and *Shroom*<sup>+/Gt</sup> E18.5 embryos and 27 for *Shroom3*<sup>Gt/Gt</sup> E18.5 embryos. n = 23 for *Nkx2–5-Cre;Shroom3*<sup>fl/fl</sup> and *Shroom3*<sup>fl/fl</sup> neonate mice and 8 for *Nkx2–5-Cre;Shroom3*<sup>+/fl</sup> neonate mice.

To measure ventricular thickness in transverse sections, 10 measurements were taken per section on 3 sequential sections per sample, each 5 µm apart. Compact layer was defined as the region between the outside of the epicardium and beginning of the trabecular zone. In example images the compact myocardial layer is delineated from the trabeculae with a dotted line. Only sections where all four chambers were visible in transverse sectioned were used for embryonic compact layer measurements. n = 6 for *Shroom3*<sup>Gt/Gt</sup>, *Shroom3*<sup>+/Gt</sup>, and *Shroom3*<sup>+/+</sup> E18.5 embryos, and n = 3 for *Shroom3*<sup>+/Gt</sup> and *Shroom3*<sup>+/+</sup> adult hearts.

To measure ventricular thickness in frontal sections, measurements were taken from 4 sequential sections, each 10 µm apart, providing a 30 µm area of average ventricle thickness. This was repeated and averaged from 4 locations within the middle of the ventricle. In example images the compact myocardial layer is delineated from the trabeculae with a dotted line. Only heart sections in which the interior of the entire ventricle was visible were used. n = 6 for *Nkx2–5-Cre;Shroom3*<sup>fl/fl</sup>, *Nkx2–5-Cre;Shroom3*<sup>+/fl</sup> and *Shroom3*<sup>fl/fl</sup> neonate hearts and adult hearts.

## Immunofluorescence

Deparaffinized tissue sections underwent antigen retrieval in a 95°C bath of sodium citrate buffer (10mM sodium citrate, 0.1% Triton-X, pH 6.0) for 25 minutes. Sections were then permeabilized with 0.2% TBST (Tween-20) for 10 minutes. Tissue sections were blocked in 10% Goat serum in 1% BSA in TBST for 1 hour. Wheat germ agglutinin Alexa Fluor 594 conjugate (5 µg/ml; Invitrogen) was used to visualize myocardial cell borders. Slides were counterstained with DAPI (1:1000, Invitrogen), cover slipped with PermaFluor Aqueous Mounting Medium (Fisher Scientific) and stored in the dark at −20°C.

## Cardiomyocyte area measurements

Adult mouse hearts were stained with Wheat germ agglutinin Alexa Fluor 594 conjugate at P8m. Cardiomyocyte cell area was measured only if cells had completely labeled membranes and a centrally located nucleus. To standardize where cardiomyocyte measurements were taken, only sections where the papillary muscle was visible were used. n = 3 for *Shroom3*$^{+/Gt}$ and *Shroom3*$^{+/+}$ hearts and n = 3 for *Nkx2–5-Cre;Shroom3*$^{fl/fl}$, *Nkx2–5-Cre;Shroom3*$^{+/fl}$ and *Shroom3*$^{fl/fl}$ hearts.

## Statistical analysis

One way ANOVA was used to assess the statistical significance of compact layer thickness differences in *Shroom3*$^{Gt}$ embryos, with a post-hoc Tukey's test. $P < 0.05$ was considered significant. An unpaired two-tailed t-test was used to determine statistical significance for compact layer and cross-sectional area measurements in adult *Shroom3*$^{Gt}$ mice. $P < 0.05$ was considered significant. One way ANOVA was used to assess the statistical significance of the heart weight, body weight, and heart weight to body weight ratios in the *Shroom3*$^{fl}$ adult mice, as well as the cross-sectional measurements of adult *Shroom3*$^{fl}$ cardiomyocytes, and of the compact layer thickness differences in *Shroom3*$^{fl}$ neonates. All statistical analysis performed using GraphPad Prism 8.0 software.

# Results

### *Shroom3* is expressed throughout the embryonic and adult myocardium

The expression range of *Shroom3* was assessed using the reporter lacZ gene found within the *Shroom3*$^{Gt}$ cassette. Whole embryos were used from E9.5 – E13.5 and excised hearts were used from E14.5 into adulthood. Wild type littermate controls were stained in parallel.

Using the reporter gene, *Shroom3* expression was detected in wholemount embryonic hearts beginning at E10.5 (Fig 1A, right) and was observed at all subsequent developmental stages including E18.5 (Fig 1I, right). Representative images for each embryonic day between these time-points are pictured in Fig 1. This expression was specific to the myocardium of the atria and ventricles of the heart and was not seen in the outflow tracts or great arteries. X-gal staining intensity increased as development progressed. When assessing sectioned embryonic hearts, *Shroom3* expression was seen earlier in development at E9.5 within the compact myocardium (Fig 2A). At E14.5 staining was seen in the trabeculae and septum, and the myocardium surrounding the base of the outflow tracts (Fig 2B–D). *Shroom3* expression was not observed in the epicardium, endocardial cushions or aortic valves at any timepoint.

In wholemount postnatal hearts, *Shroom3* expression was also seen in the myocardium of the ventricles and atria, and the myocardium surrounding the base of the outflow tracts. The widespread, high intensity staining which is seen in newborn hearts (Fig 3A) is similar to staining seen in late-stage embryonic hearts. However, this staining intensity dramatically decreases by three months of age (Fig 3B) and eight months of age (Fig 3C). Notably, the staining intensity of the left atrium remained strong in comparison to the right atrium in adulthood (Fig 3C). Assessing sectioned adult hearts, *Shroom3* expression was again specific to the compact and trabecular myocardium and absent from the epicardium and endocardium, with increased staining in the compact layer (Fig 4A, B). This is a similar pattern to that seen in embryonic

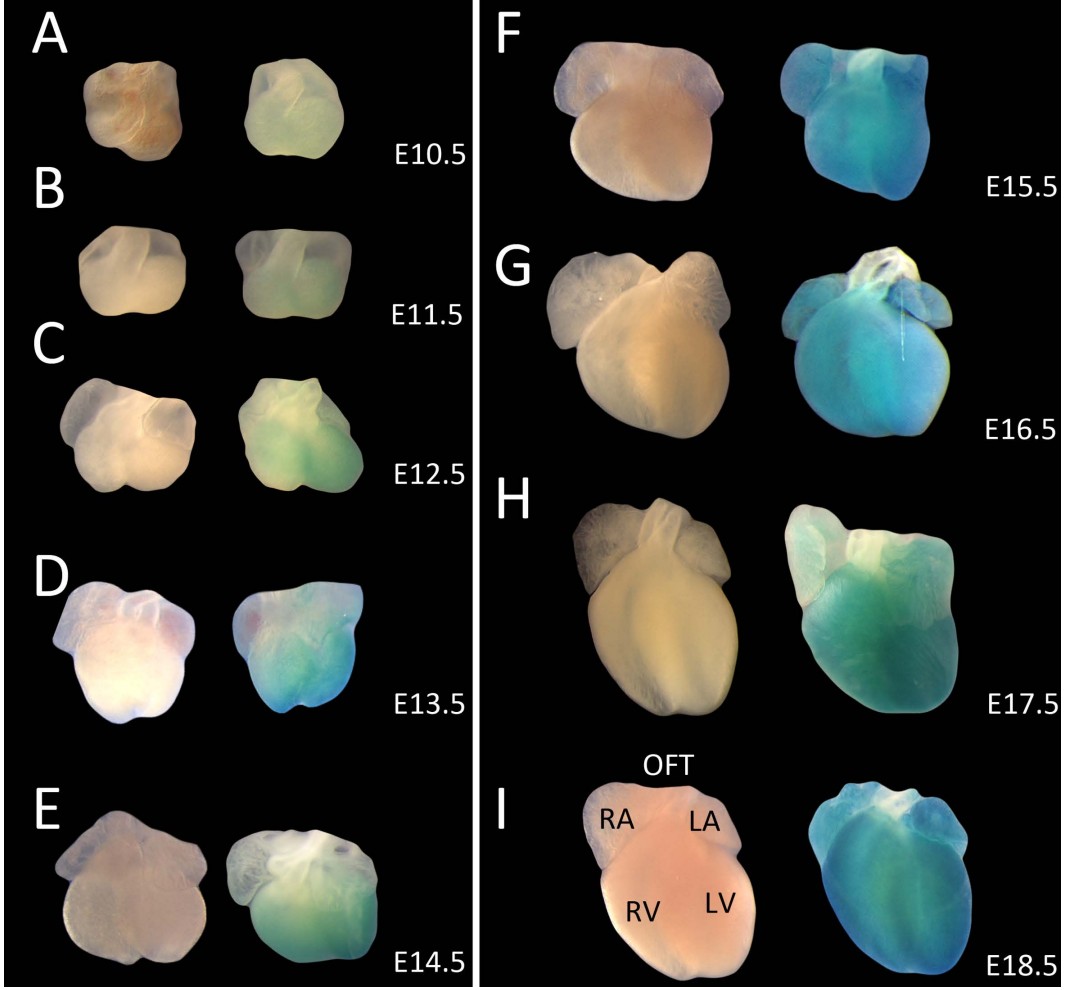

**Fig 1. _Shroom3_ is expressed in the mouse heart during cardiogenesis.** X-gal staining in embryonic mouse hearts detecting for presence of lacZ in _Shroom3_^+/Gt embryos (right), or littermate controls (left). Wholemount detection began at E10.5 (A) and was seen for the remainder of the gestational period, until E18.5 (B-I), in a widespread and consistent pattern. Detection was seen in the atria and ventricles, but not in the outflow tracts (A-I). Wild type control hearts stained in parallel showed no background staining. LA: left atrium; LV: left ventricle; OFT: outflow tracts; RA: right atrium; RV: right ventricle.

heart sections. Reflecting the findings in wholemount hearts, increased staining in the left atrium compared to the right atrium was also observed in section (Fig 4C, D).

**Full-body loss of _Shroom3_ during development shows congenital heart defects in embryos and ventricular thinning and decreased cardiomyocyte size in adults**

As _Shroom3_ was seen in much of the developing heart and was present for the majority of cardiogenesis, embryos were assessed at the end of development for CHDs after full-body developmental _Shroom3_ loss. In E18.5 embryos three types of CHDs were found. Firstly, 55% of _Shroom3_^Gt/Gt embryos contained VSDs (18 of 33 embryos). Of these VSDs, 72% were membranous, occurring in the first third of the ventricles (Fig 5B, C), and 28% were muscular, occurring lower in the muscular septum (Fig 5E, F). Secondly, thinning of the left ventricle wall was also found in these embryonic hearts,

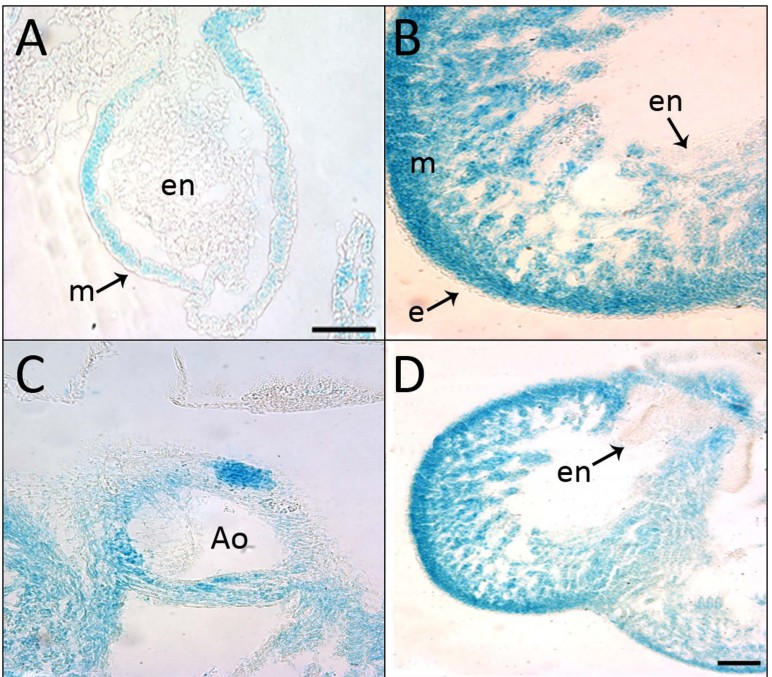

**Fig 2. *Shroom3* is expressed throughout the mouse myocardium during cardiogenesis.** X-gal staining in paraffin sections of *Shroom3*+/Gt embry-onic heart. A) The primitive ventricle of an E9.5 mouse embryo showed presence of lacZ in the myocardium (arrow) but not the endocardium. B-D) Hearts sections of an E14.5 embryo showed widespread staining in the compact layer myocardium, trabecular myocardium, ventricular septum and the base of the aorta. No staining was detected in the endocardium, epicardium (arrows), aortic valve, or endocardial cushions. Ao: aorta; e: epicardium; en: endocardium; m: myocardium.

where a significant, stepwise reduction in compact myocardium thickness was seen between WT, *Shroom3*+/Gt, *Shroom3*Gt/Gt hearts (Fig 6C). The compact myocardium was defined as the layer between the epicardium and the trabecular myo-cardium boundary (Fig 6B, dotted line). No significant difference in compact myocardium thickness was seen between genotypes in the right ventricle. The morphology of the trabeculae networks was also assessed at this time, however there was no difference seen in network branching between genotypes. Finally, abnormal aortic valves were seen in 15% of *Shroom3*Gt/Gt embryos (4 of 27 embryos), and abnormal pulmonary valves were seen in 26% of *Shroom3*Gt/Gt embryos (7 of 27 embryos). This appeared as aortic valves with no clear delineation of the leaflet cusps (Fig 7C and D), and pulmonary valves where leaflet cusps were clustered within the valve lumen (Fig 7G and H). In all embryonic hearts, there was no observed incidence of cardia bifida, failure to loop or form chambers, failure to trabeculate, nor hypertrophy. As seen in the representative images in Fig 2, normal gross morphology was observed for all hearts at all developmental time points, where no obvious alterations in overall size and shape were seen.

Due to lethal extra-cardiac developmental defects, *Shroom3*Gt/Gt embryos are not able to survive into adulthood. How-ever, *Shroom3*+/Gt mice can survive past weaning and into adulthood. Therefore, mice heterozygous for full-body *Shroom3* loss were assessed for long-term consequences of left ventricular thinning during development. At three months postnatal, *Shroom3*Gt/+ hearts showed significantly reduced left ventricular wall thickness compared to WT controls (Fig 8A–C). This was also seen at eight months postnatal (Fig 8D–F). As ventricular wall thinning may result in stress on the ventricles and thus cause compensatory cardiomyocyte hypertrophy, we also assessed cardiomyocytes at eight months postnatal. Using wheat germ agglutinin to outline cardiomyocyte cell walls, surface areas of cardiomyocytes from the left ventricle compact

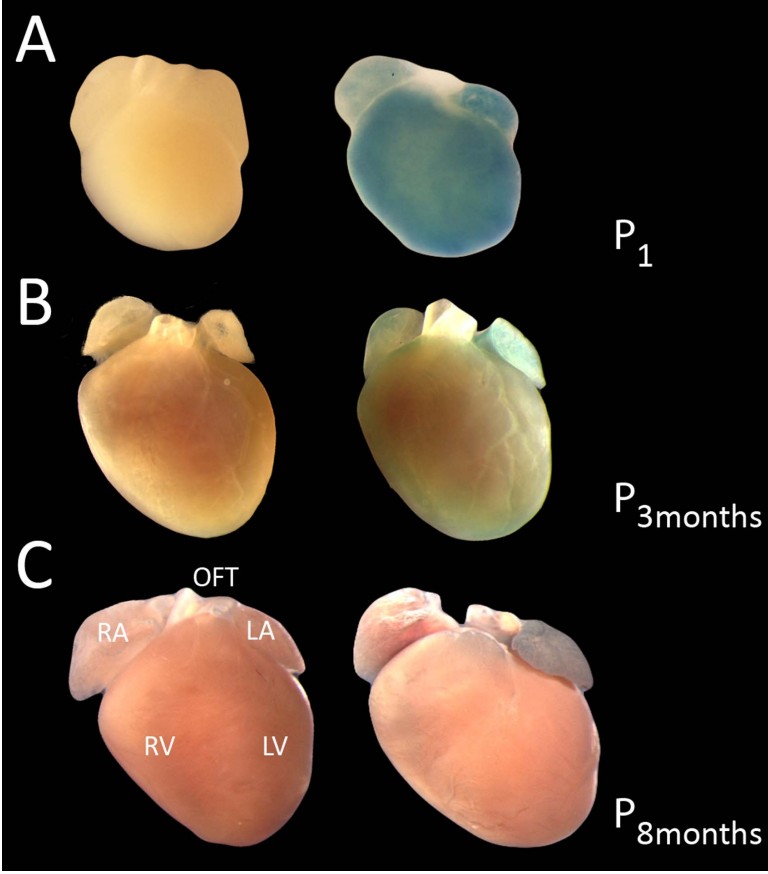

**Fig 3. *Shroom3* is expressed in the postnatal mouse heart.** X-gal staining in postnatal mouse hearts detecting for presence of lacZ in *Shroom3+/Gt* (right), or littermate controls (left). Wholemount detection began on the day of birth (A), where an expression pattern similar to embryonic hearts was observed. Expression of lacZ in 3-month-old (B) and 8-month-old (C) mice, respectively, showed increased staining in the left atrium compared to the right atrium. Widespread expression throughout the ventricles appeared to be consistent with embryonic expression, however this was difficult to observe in wholemount. Expression was observed in the myocardium surrounding the base of the outflow tracts. Wild type control hearts stained in parallel showed no background staining. LA: left atrium; LV: left ventricle; OFT: outflow tracts; RA: right atrium; RV: right ventricle.

myocardium were measured. Cardiomyocytes from *Shroom3Gt/+* left ventricles were found to be significantly smaller than the WT controls (Fig 9).

## Floxed allele recombination and verification

Following this data, we aimed to investigate if these developmental defects were due to *Shroom3* loss in the myocardium during development. Our lab and two collaborating labs funded the creation of a novel floxed *Shroom3* allele (*Shroom3fl*). Placement of the loxP sites was designed to splice around exon 5, creating a null *Shroom3* allele which does not express *Shroom3* mRNA. This line has recently been published by our collaborators to study optic cup morphogenesis [28].

The specificity of the loxP targeting was assessed by making a compound allele. The *Shroom3fl* line was crossed with the *Shroom3Gt* line, resulting in embryos containing the compound *Shroom3Gt/fl* alleles. This would create a null allele from the gene trap insertion and a null allele from recombination of the floxed *Shroom3* allele, driven by the Cre recombinase inside the *Shroom3Gt* cassette. Compound allele embryos phenocopied *Shroom3Gt/Gt* embryos at E10.5, where similar

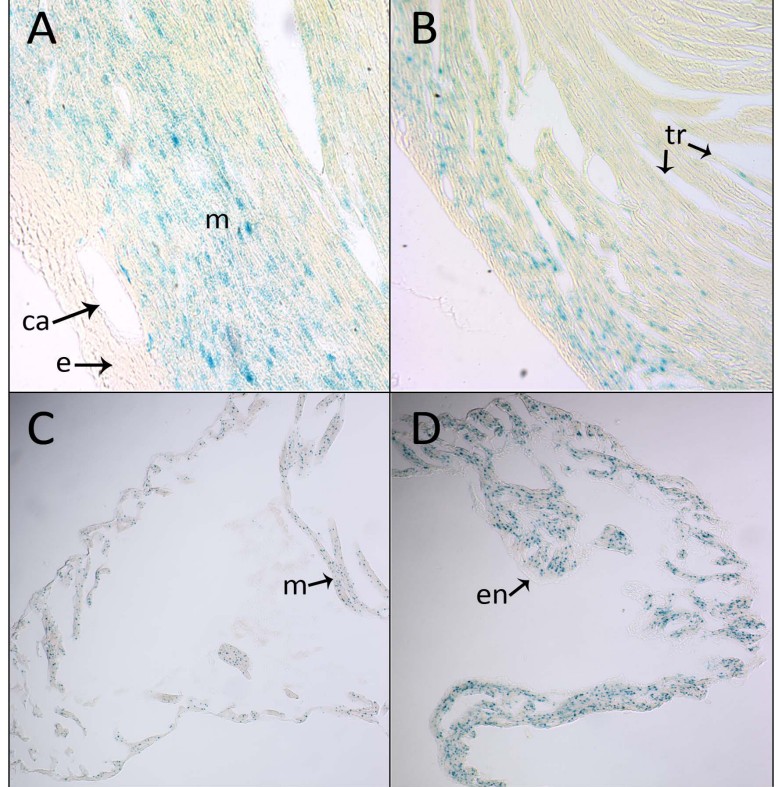

**Fig 4. *Shroom3* is expressed throughout the mouse myocardium postnatally.** X-gal staining in paraffin sections of *Shroom3*[+/Gt] 8-month-old adult heart. Transverse sections through the left (A) and right (B) ventricle showed intense staining in the compact layer myocardium and to a lesser extent the trabeculae. No staining was seen in the epicardium. Sagittal sections through the right (C) and left (D) atrium showed increased staining intensity in the left atrium. No staining was found within the endocardium or epicardium. ca: coronary artery; e: epicardium; en: endocardium; m: myocardium; tr: trabeculae.

neural tube closure defects were seen (Fig 10A and B). The same phenocopying was seen at E18.5, where the same exencephaly, gut tube looping, and open eye phenotypes were seen (Fig 10C and D). Using similar methods, our collaborators have obtained similar results at E12.5 [28].

To target *Shroom3* loss to the developing myocardium, *Shroom3*[fl] mice were crossed to an *Nkx2–5-Cre* recombinase line. This transgenic Cre recombinase begins expression at E7.5 in cardiac precursor cells, and expression remains on for the duration of development and into adult life [35]. Primers to amplify exon 5 of the conditional KO *Shroom3*[fl] allele, and the recombined *Shroom3*[fl] allele were used with cDNA. Anticipated band sizes for the conditional KO allele (181 bp) and the recombined allele (306 bp) were seen in agarose gel in both neonate and adult heart samples (S2A Fig). We also verified that this loss was specific to the heart by using kidney samples from the same mice. cDNA samples from neonate control hearts and neonate knockout hearts (n = 4) were then amplified for the conditional KO allele or the recombined allele (S2B Fig). Band densitometry was used to quantify recombination efficacy in the *Nkx2–5-Cre;Shroom3*[fl] hearts. The average difference in band density between *Shroom3*[fl] littermate controls and *Nkx2–5-Cre;Shroom3*[fl] hearts was 74.5% (S2C Fig). This indicates at least a 74% difference in *Shroom3* expression between *Nkx2–5-Cre;Shroom3*[fl] mice and littermate controls, or a 74% knockout of the *Shroom3*[fl] allele from our myocardial specific Cre driver.

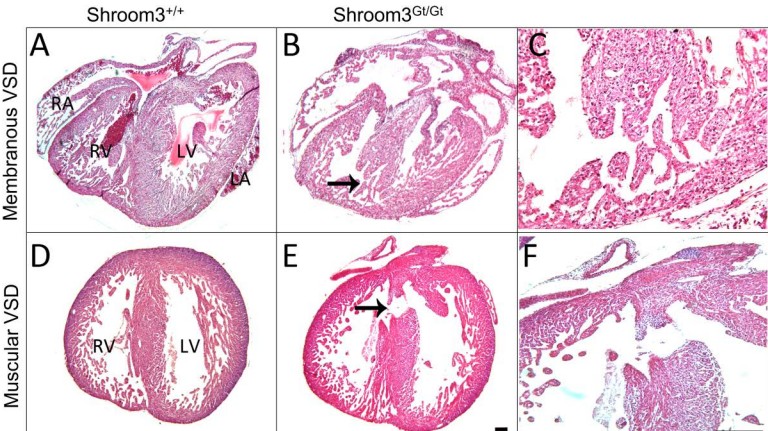

**Fig 5. Embryonic *Shroom3^Gt/Gt* hearts show ventricular septal defects.** H&E staining in E18.5 hearts. A&D) Transverse sections of wild type hearts show an intact ventricular septum between the two ventricles, at two different heights of the heart. B) Transverse section of a *Shroom3^Gt/Gt* heart with a membranous VSD (black arrow). C) Magnification shows an otherwise intact tissue morphology. E) Transverse section of a *Shroom3^Gt/Gt* heart with a muscular VSD (black arrow). F) Magnification shows an otherwise intact tissue morphology. Incidence: *Shroom3^+/+* = 0/24; *Shroom3^+/Gt* = 1/24; *Shroom3^Gt/Gt* = 18/33. LA: left atrium; LV: left ventricle; RA: right atrium; RV: right ventricle.

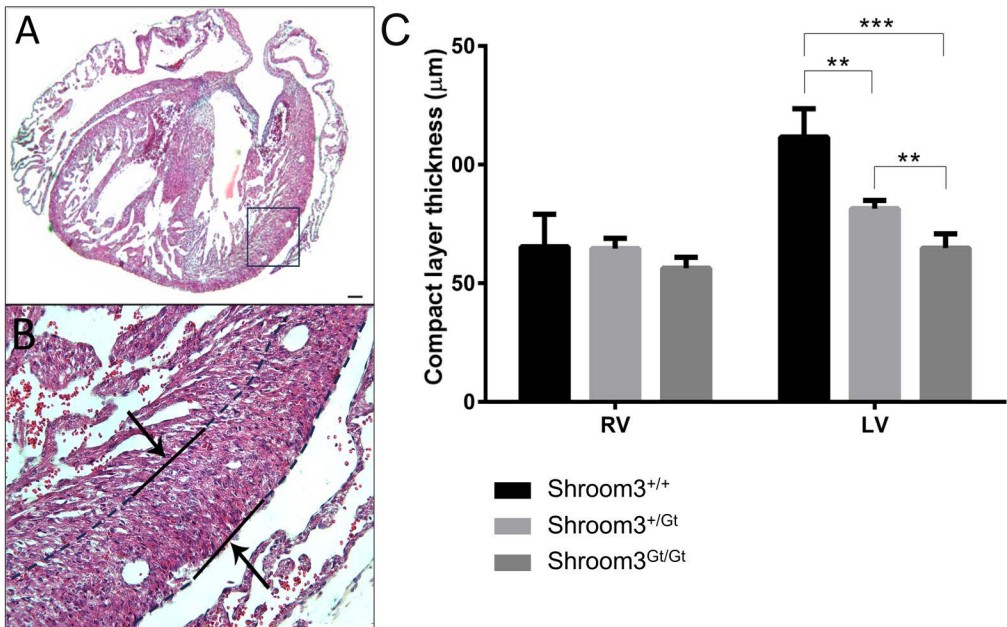

**Fig 6. Embryonic *Shroom3^Gt/Gt* hearts show left ventricular thinning.** Measurements of the compact layer thickness in the wall of the right and left ventricle of E18.5 embryos. The compact layer was defined as the space between the outer epicardium and where the trabeculae begin. A&B) Transverse section demonstrating where the compact layer measurements were taken for each heart section. Lines in B indicate the border of the compact layer. C) Quantification of the compact layer width showed no significant differences in the right ventricle across all genotypes. In the left ventricle, compact layer thickness was significantly reduced in *Shroom3^+/Gt* ($P < 0.01$) and *Shroom3^Gt/Gt* ($P < 0.001$) hearts compared to littermate controls. Additionally, *Shroom3^Gt/Gt* left ventricle compact layer thickness was significantly thinner than *Shroom3^+/Gt* ($P < 0.01$). ***$P < 0.001$, **$P < 0.01$. ±SD. RV: right ventricle; LV: left ventricle. n = 6 per genotype.

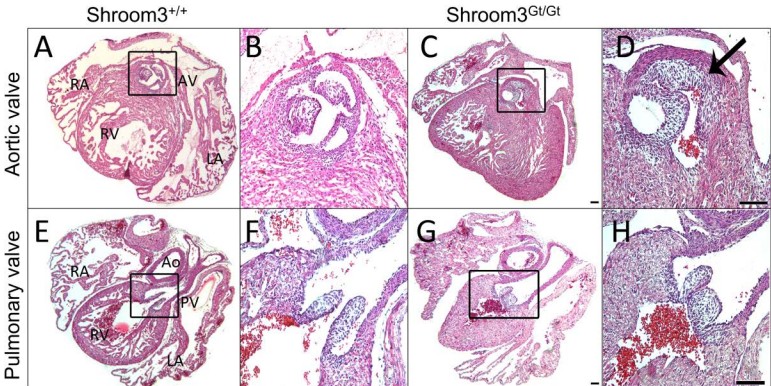

**Fig 7. Embryonic Shroom3<sup>Gt/Gt</sup> hearts show abnormal semilunar valves.** H&E staining in E18.5 hearts. Assessment of phenotype was based on comparison to the littermate controls. A&E) Transverse sections of wild type hearts with normal development of the aortic (A) and pulmonary (E) valve. Magnifications to the right (B&F). C) Transverse section of *Shroom3<sup>Gt/Gt</sup>* heart at the site of the aortic valve with undefined cusp formation. D) Magnification of C. Compared to the control valve, leaflets of the aortic valve in *Shroom3<sup>Gt/Gt</sup>* hearts are not clearly defined (black arrow) and do not extend into the luminal space of the valve. Incidence: *Shroom3<sup>+/+</sup>* = 0/18; *Shroom3<sup>+/Gt</sup>* = 0/18; *Shroom3<sup>Gt/Gt</sup>* = 4/27. G) Transverse section of *Shroom3<sup>Gt/Gt</sup>* heart at the site of the pulmonary valve with thick pulmonary cusps. (H) Magnification of G. From visual comparison, pulmonary valves in *Shroom3<sup>Gt/Gt</sup>* hearts were thickened compared to control valves. Incidence: *Shroom3<sup>+/+</sup>* = 0/18; *Shroom3<sup>+/Gt</sup>* = 1/18; *Shroom3<sup>Gt/Gt</sup>* = 7/27. Ao: aorta; AV: aortic valve; LA: left atrium; LV: left ventricle; PV: pulmonary valve; RA: right atrium; RV: right ventricle.

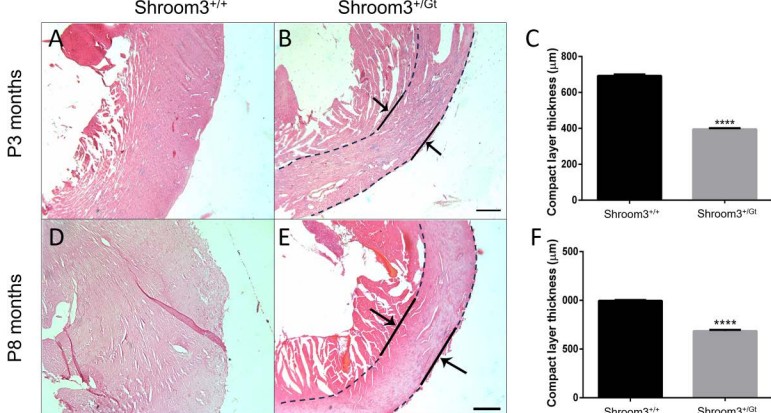

**Fig 8. Adult Shroom3<sup>+/Gt</sup> mouse hearts show left ventricular thinning.** Transverse sections of adult mouse hearts stained with hematoxylin and eosin. Only the compact layer of the left ventricle was compared between genotypes for thinning. The compact layer was defined as the space between the outside of the epicardium and where the trabeculae begin. A&B) Wild type and *Shroom3<sup>+/Gt</sup>* 3-month-old hearts. Arrows and lines indicate where measurements in the left ventricle were taken. C) Quantification of the thickness of the compact layers of 3-month-old hearts showed a significant decrease in left ventricle wall thickness in the heterozygous mice (P<0.0001) when compared to the wild-type. D&E) Transverse sections of wild type and *Shroom3<sup>+/Gt</sup>* 8-month-old hearts. Arrows and lines indicate where measurements in the left ventricle were taken. F) Quantification of the thickness of the compact layers of 8-month-old hearts showed a significant decrease in left ventricle wall thickness in the heterozygous mice (P<0.0001) when compared to the wild type. ****P<0.0001. ±SD. RV: right ventricle; LV: left ventricle. n=3 per genotype at both ages. Mixed cohorts used at 3 months and 8 months.

## Myocardial *Shroom3* loss during development does not produce congenital heart defects in embryos and does not impact adult cardiac or cardiomyocyte morphology

To assess for perinatal lethality in our myocardial specific knockout, four *Nkx2–5-Cre;Shroom3<sup>+/fl</sup>* X *Shroom3<sup>fl/fl</sup>* breeding pairs were set up, two with paternally inherited Cre, two with maternally inherited Cre. Neonates born from these crosses

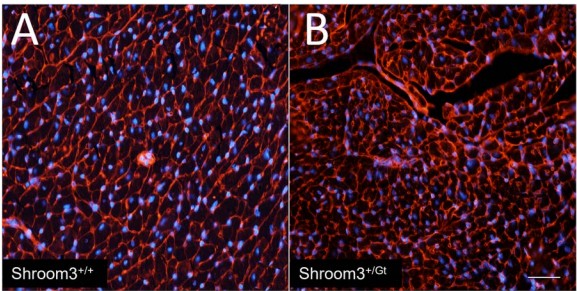

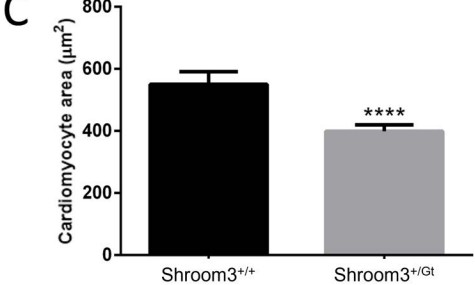

**Fig 9. Adult *Shroom3*+/Gt hearts show decreased cardiomyocyte cross-sectional area.** Hearts from 8-month-old mice were stained with wheat germ agglutinin conjugated to Alexa Fluor 594 (red) to mark the cell membranes and DAPI (blue). Only cells with fully defined and intact cell membranes and a centralized nucleus were used for measurements. A) Wild type heart section showing normal cardiomyocytes in cross section. B) *Shroom3*+/Gt heart cardiomyocytes in cross section. C) Hearts of heterozygous mice showed a significantly smaller cross-sectional area (P < 0.0001) when compared to the wild type. ****P < 0.0001. ± SD. n = 3 per genotype. Mixed cohort used.

were monitored for the first 12 hours of life. All observed neonates were born alive, however any deaths within the first 12 hours of life were recorded and bodies were collected. From a final n-value of 44, 21 neonates were born with the Cre-recombinase, and 23 were born without. Within the first 12 hours of life, 5 neonates from both groups died. This indicates a mortality rate of 23% across *Nkx2–5-Cre;Shroom3*+/fl and *Nkx2–5-Cre;Shroom3*fl/fl neonates, and 21% across *Shroom3*+/fl and *Shroom3*+/fl neonates. Overall, no trend in mortality was seen across genotypes in neonates. Additionally, observed genotypes across all neonates did not differ from expected mendelian ratios ($X^2 = 0.72$, df = 3, p = 0.05), indicating no significant *in-utero* mortality.

Neonates from this cross were assessed for CHDs using the same methods as in *Shroom3*Gt embryos. In both *Nkx2–5-Cre;Shroom3*+/fl and *Nkx2–5-Cre;Shroom3*fl/fl neonate hearts, no VSDs were seen (Fig 11). This includes neither membranous nor muscular VSDs, both of which had been seen in *Shroom3*Gt/Gt embryos. Additionally, in comparing to littermate controls, aortic and pulmonary valves were morphologically normal regardless of phenotype (Fig 12). Additional images of these valves are displayed in S3 Fig. Finally, no significant differences were seen in the ventricular compact layer thickness between *Nkx2–5-Cre;Shroom3*+/fl, *Nkx2–5-Cre;Shroom3*fl/fl neonates and littermate controls (Fig 13). This was consistent in both the right (p = 0.1522) and left ventricle (p = 0.8859) (Fig 13C).

Mice born from this cross were also monitored for a year with no notable issues in overall health or breeding. At one year of age no alterations to gross morphology in adult hearts were observed. This includes no minor VSDs which persisted into adulthood and no alterations to ventricular wall thickness (Fig 14A, first and second column). Additionally, there were no morphological alterations in the semilunar valves of these hearts (Fig 14A, third and fourth column). No alterations in body weight (Fig 14B, left) or heart weight (Fig 14B, middle) were seen in these mice, reflecting no changes in heart weight to body weight ratios (Fig 14B, right). Thus, there is no indication of compensatory cardiac hypertrophy after one year of myocardial *Shroom3* loss.

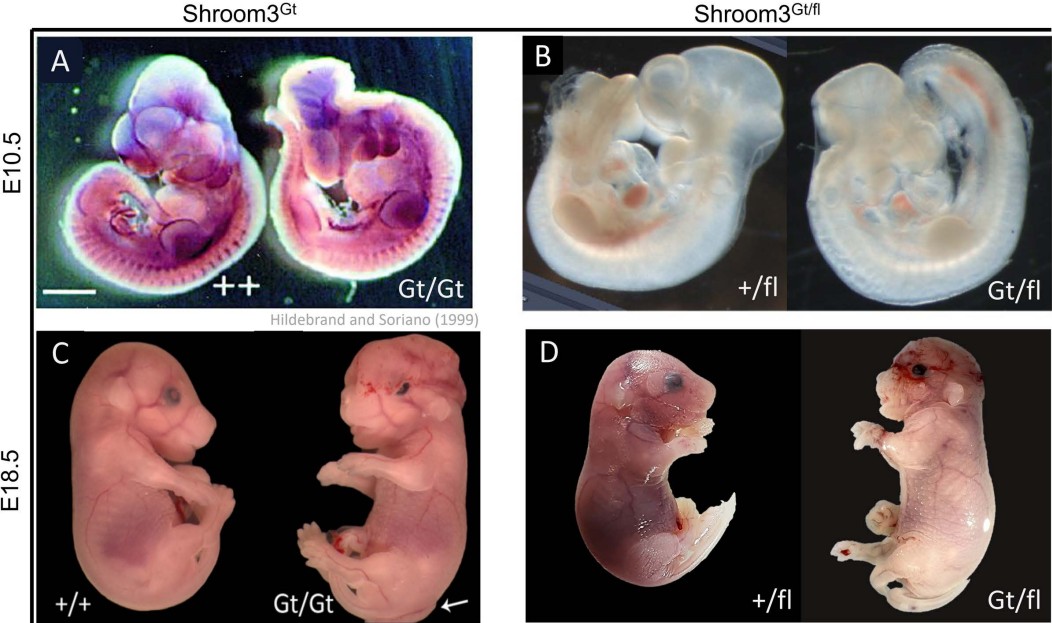

**Fig 10. Embryos with compound _Shroom3^(Gt/fl)_ alleles phenocopy _Shroom3^(Gt/Gt)_ embryos.** A) Full-body _Shroom3_ loss using the gene trap line, visualized at E10.5. Homozygous _Shroom3_ loss on the right, and littermate control on the left. On the right, neural tube closure defects can be seen. B) Embryos from a _Shroom3^(Gt/+)_ X _Shroom3^(fl/fl)_ cross at E10.5. 8 embryos were produced. 4 showed the WT phenotype (left) and 4 showed the mutant phenotype (right). This mutant phenotype seen in A, right. Mating cross and data collection by T.J Plageman. C) Full-body _Shroom3_ loss using the gene trap line, visualized at E18.5. Homozygous _Shroom3_ loss on the right, and littermate control on the left. On the right, neural tube defects, eye defects, and gut tube looping defects can be seen. D) E18.5 embryos from _Shroom3^(Gt/+)_ X _Shroom3^(fl/fl)_. 6 embryos produced, 3 with WT phenotype (left), 3 with neural tube defects, open eye phenotype, gut tube looping defects (right).

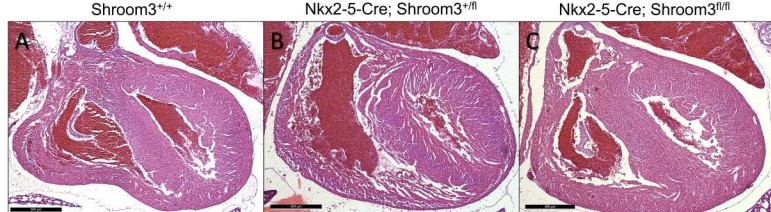

**Fig 11. Myocardial _Shroom3_ loss during development does not show ventricular septal defects.** Neonates from the _Nkx2-5-Cre;Shroom3^(+/fl)_ X _Shroom3^(fl/fl)_ cross who died within 24 hours of birth display no VSDs. This was consistent between littermate neonates which did not contain Cre (A), littermates heterozygous for _Shroom3_ recombination (B), and littermates homozygous for _Shroom3_ recombination (C). n = 23 for _Nkx2-5-Cre;Shroom3^(fl/fl)_ and _Shroom3^(fl/fl)_. n = 8 for _Nkx2-5-Cre;Shroom3^(+/fl)_. Sectioned in the frontal plane.

Adult hearts were also assessed at 8 months of age for any changes in cardiomyocyte morphology using the same wheat germ agglutinin staining as in _Shroom3^(Gt)_ adults. The average cardiomyocyte area between littermate controls, _Nkx–5-Cre;Shroom3^(+/fl)_, and _Nkx–5-Cre;Shroom3^(fl/fl)_ mice were not significantly different (p = 0.1956) (Fig 15).

## Discussion

### _Shroom3_ expression begins early in heart development and continues in the adult heart

We aimed to assess when expression of _Shroom3_ began during heart development, if it was sustained during and after development, and where this expression was localized. Through X-gal staining, we have demonstrated that _Shroom3_

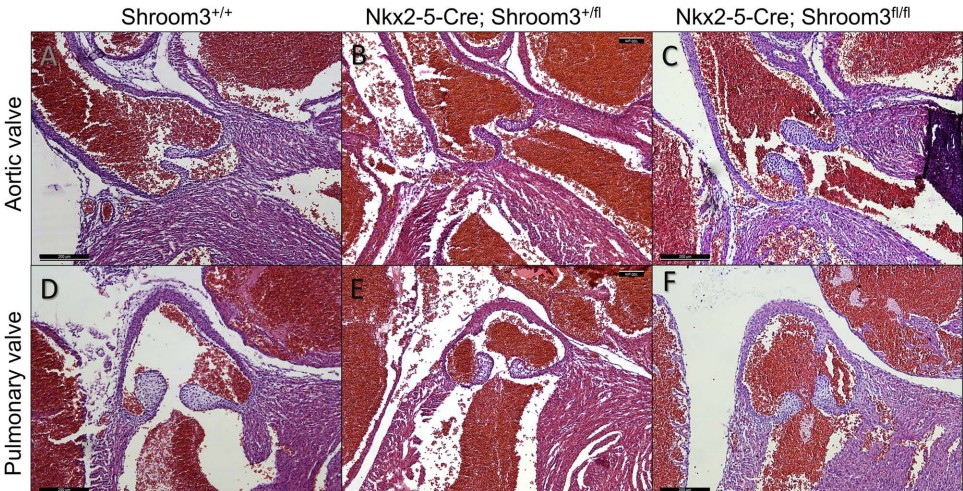

**Fig 12. Myocardial *Shroom3* loss during development does not show abnormal semilunar valves.** Neonates from the *Nkx2-5-Cre;Shroom3*⁺/ᶠˡ X *Shroom3*ᶠˡ/ᶠˡ cross show normal pulmonary and aortic valve leaflets compared to littermate controls. This was consistent between littermate neonates which did not contain Cre (A), littermates heterozygous for *Shroom3* recombination (B), and littermates homozygous for *Shroom3* recombination (C). n = 23 for *Nkx2-5-Cre;Shroom3*ᶠˡ/ᶠˡ and *Shroom3*ᶠˡ/ᶠˡ. n = 8 for *Nkx2-5-Cre;Shroom3*⁺/ᶠˡ. Sectioned in the frontal plane.

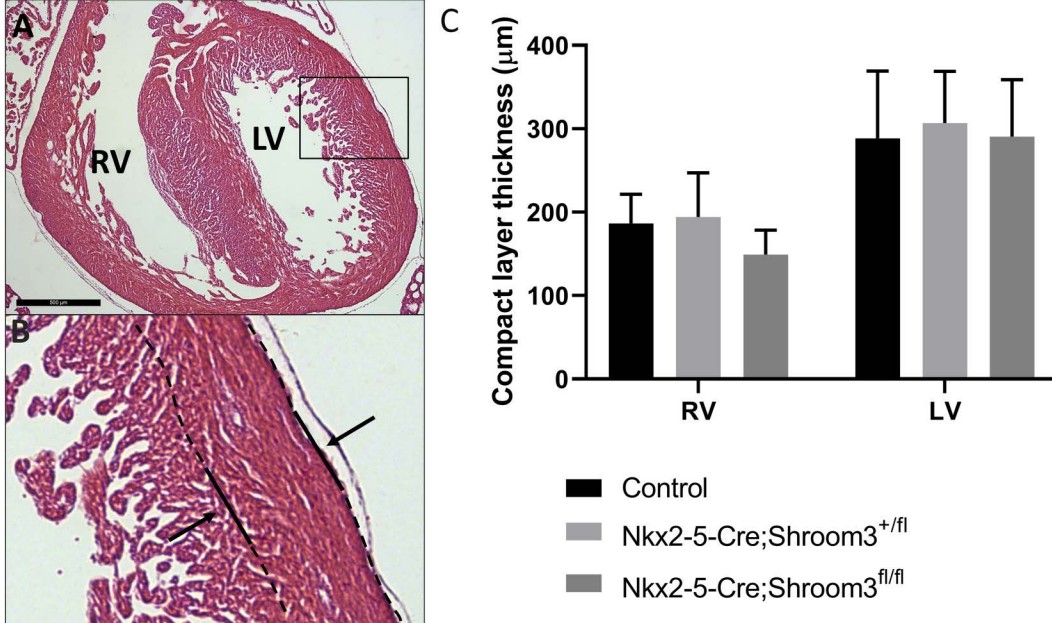

**Fig 13. Myocardial *Shroom3* loss during development does not alter ventricular wall thickness.** Measurements of compact myocardium were taken from *Nkx2-5-Cre;Shroom3*⁺/ᶠˡ X *Shroom3*ᶠˡ/ᶠˡ neonate litters. B) Measurements were taken between the two lines, for both the left and right ventricle. C) No significant differences were found in wall thickness between the different genotypes for both the right ventricle (p = 0.1522) and left ventricle (p = 0.8859). Control label indicates *Shroom3*⁺/ᶠˡ and *Shroom3*ᶠˡ/ᶠˡ samples. One way ANOVA ±SD. n = 6 per genotype.

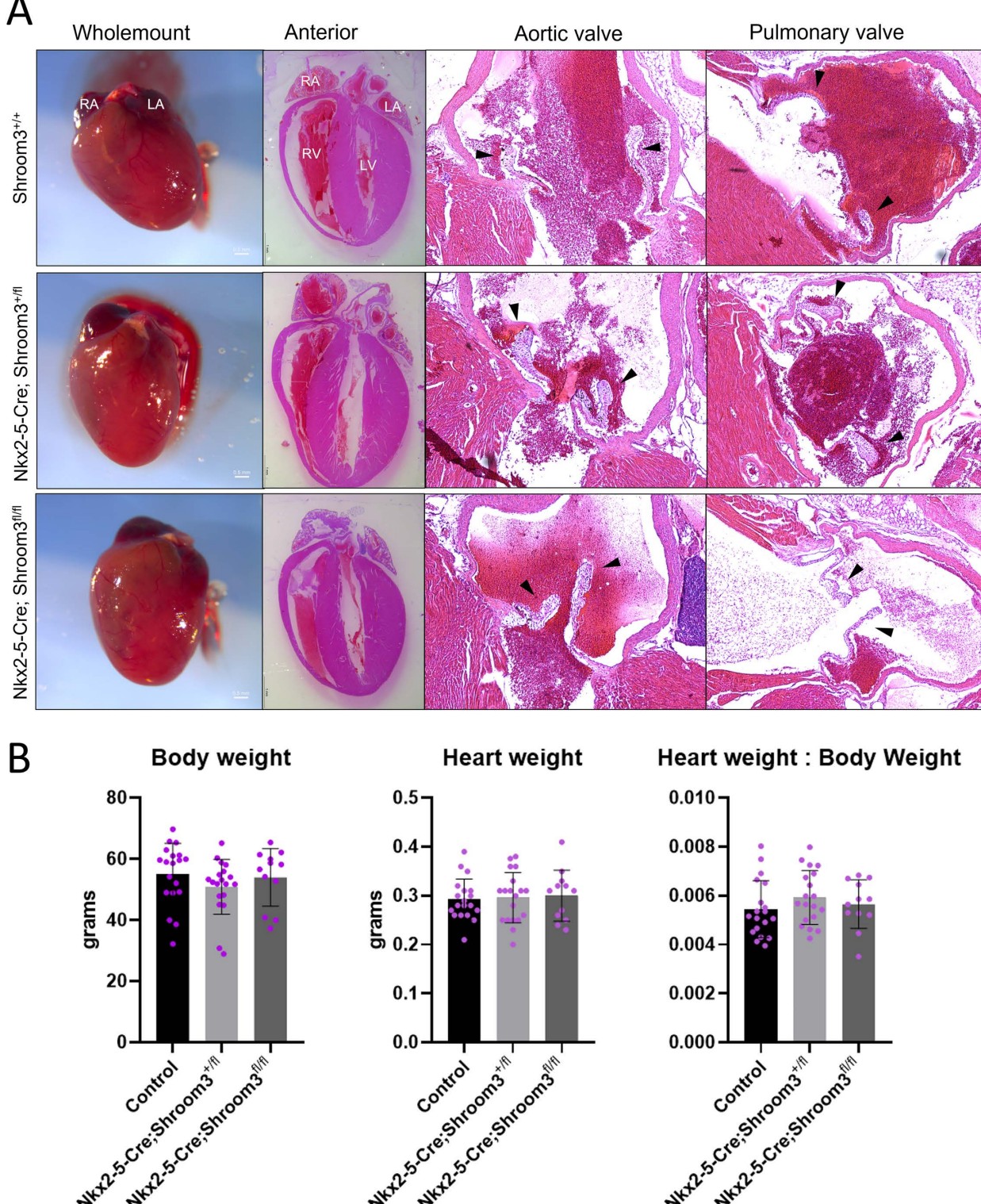

**Fig 14. *Shroom3* myocardial specific loss does not alter postnatal cardiac morphology.** A) Representative images of 1y adult hearts from myocardial *Shroom3* loss during development. *Shroom3fl*, *Nkx2-5-Cre;Shroom3+/fl*, and *Nkx2-5-Cre;Shroom3fl/fl* mice were observed to show no changes in heart shape or size externally and internally, with no changes in ventricular thickness, and no alterations to the semilunar valves. All hearts were equal in appearance. B) Body weight (left), heart weight (middle), and heart weight to body weight ratios (right) from *Nkx2-5-Cre;Shroom3+/fl* X *Shroom3fl/fl*

offspring at 1y, by genotype. No statistical differences seen (left, p = 0.3490; middle, p = 0.9156; right, p = 0.4829). Control label indicates *Shroom3*<sup></sup> *Shroom3*+/fl and *Shroom3*fl/fl samples. One-way ANOVA ±SD. n = 19 for *Shroom3*fl, n = 19 for *Nkx2-5-Cre;Shroom3*+/fl, n = 12 for *Nkx2-5-Cre;Shroom3*fl/fl. No significance seen in sex segregated analysis.

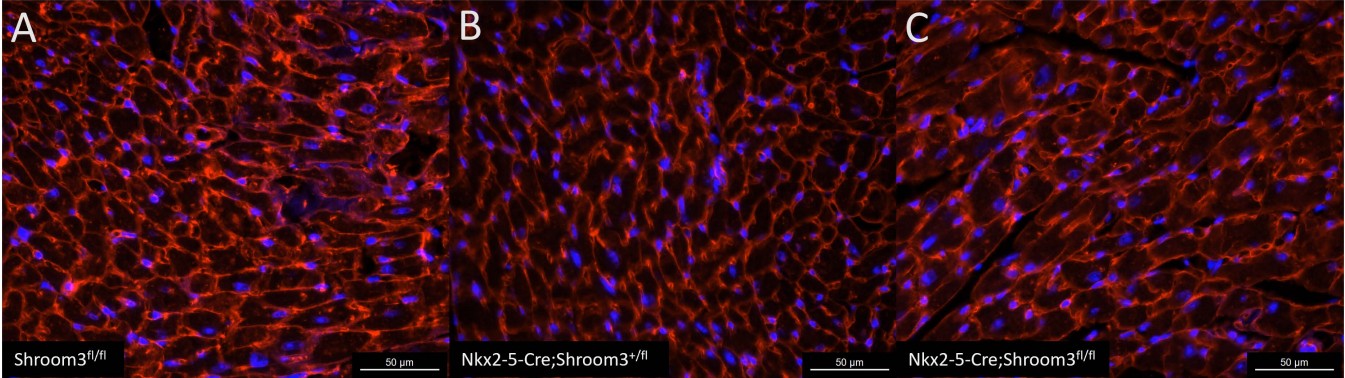

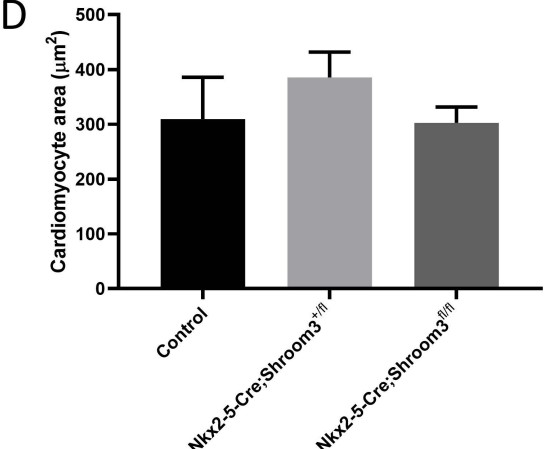

**Fig 15. Developmental loss of *Shroom3* in the myocardium does not affect cardiomyocyte cross sectional area in adults.** Hearts from 8-month-old mice were stained with wheat germ agglutinin conjugated to Alexa Fluor 594 (red) to mark the cell membranes and DAPI (blue). Only cells with well-defined and intact cell membranes and a centralized nucleus were used for measurements. A) Littermate control heart section showing normal cardiomyocytes. B) *Nkx2-5-Cre;Shroom3*+/fl heart cardiomyocytes in cross section. C) *Nkx2-5-Cre;Shroom3*fl/fl heart cardiomyocytes in cross section. D) No significant differences in cardiomyocyte cross sectional area were seen in between all three genotypes. Control label indicates *Shroom3*+/fl and *Shroom3*fl/fl samples. (p = 0.1956) ±SD. n = 3 per genotype.

is expressed in the developing heart as early as E9.5, continues for the entirety of development, and into adult life. This expression was localized to the myocardium and trabeculae of the atria and the ventricles, and myocardium surrounding the base of the outflow tracts. *Shroom3* expression was not found in the outflow tracts, epicardium, endocardium, or the valves of the developing and adult heart. Due to the increase of X-gal staining intensity over the course of development, it is likely that *Shroom3* expression is increasing in the embryonic heart as gestation progresses. However, staining intensity decreased in the adult heart. This shift in expression between embryonic and adult hearts may be due to the functionality of the heart at different stages of life. In utero, trabeculae contribute to the force of cardiac output, and act as an important biomechanical support structure for proper downstream heart development [36–38], whereas adult hearts rely on the muscular compacted myocardium in the ventricular walls.

A recent study has established spatially distinct transcriptomes in cardiomyocytes of adult mice which radiate out from an area of infarct [39]. In this study, Shroom3 was expressed in a transitional cell population found between non-ischemic tissue and cardiomyocytes within the infarct zone, but not highly expressed within either of these areas. It is known that as the heart tries to recover from ischemia, developmental programming within the cells is re-established in a compensatory, regenerative manner, for example the change from adult α-myosin heavy chain to embryonic β-myosin heavy chain [40,41]. This presents the possibility that under conditions of heart failure or ischemic damage, *Shroom3* may be upregulated in the adult heart in a compensatory manner. This supports our findings from X-gal staining that Shroom3 expression is decreased from developmental levels in normal adult mouse hearts.

In adult hearts, there was also a notable difference in X-gal staining intensity between the left and right atria, with higher *Shroom3* expression in the left atrium compared to the right. This staining pattern is similar to that of Pitx2, which also shows greater expression in the left mouse atrium than the right [42]. While *Shroom3* is activated by Pitx2 in the developing gut, this interaction has not been studied in the heart at any time-point [17,27]. Additionally, as cell shape changes like those seen during gut tube looping are not present in adult atria, the purpose for this interaction is unknown. Pitx2c is known to be involved in left-right asymmetry during embryonic development [43]. While a *Shroom3* missense mutation has been identified in a human patient with heterotaxy [7], it seems unlikely that *Shroom3* is implicated in left-right asymmetry in the heart, as no looping defects were seen in this study. Using the same *Shroom3*^Gt line Durbin *et al.*, (2020) had also investigated *Shroom3* expression patterns in the mouse heart, however they did not document differential staining of the adult atria.

## SHROOM3, but not myocardial SHROOM3, is required for normal cardiac morphogenesis

As *Shroom3* is so widespread in the developing heart, we aimed to investigate the implications of full-body *Shroom3* loss on cardiogenesis using the *Shroom3*^Gt mouse line. In E18.5 embryos, this loss resulted in thinning of the compact myocardium isolated to the left ventricle. Both membranous and muscular ventricular septal defects were also observed in these embryos, as well as malformation of the semilunar valve leaflets. These phenotypes were not completely penetrant amongst all embryos and did not always appear together in the same combinations. These results can be compared to Durbin *et al.*, (2020) who also showed ventricular septal defects and thinning of the left ventricle in E14.5 *Shroom3*^Gt/Gt embryos. However, this group did not demonstrate remodelling complications of the semilunar valves. This may be due to assessment of embryos at different developmental time-points. At E14.5, the ventricles of the mouse heart have not completed proliferation and compaction, which occurs from E15.5 until birth [44]. Downstream of this, cardiac valves, but particularly the semilunar valves, are known to be morphologically influenced by cardiac hemodynamics and pressure during development [45–47]. It is possible that valve defects from *Shroom3* loss were only seen for the first time in our study due to the influence of the hemodynamic changes from the hearts undergoing ventricular compaction at later developmental stages.

Following this, we investigated if it was SHROOM3 arising from the myocardium which, when lost, was causing these defects. However, when *Shroom3* was eliminated using an *Nkx2–5* promoter driven Cre recombinase, no abnormal phenotypes were seen in neonate or adult hearts. This included no VSDs, valve deformities, and no ventricular thinning.

With this data, we have demonstrated that *Shroom3* loss does cause CHDs, but these defects are not due to loss of *Shroom3* in the myocardium. Thus, we hypothesize that the defects seen in the *Shroom3*^Gt knockout must be due to loss of *Shroom3* in extracardiac cell populations which also contribute to cardiac development. The best candidate for this is the cardiac neural crest cell (cNCCs). cNCCs are a migratory cell population which arise dorsal to the neural tube. They have been well established in the formation of the great arteries, including the patterning and smooth muscle of the aorta, and septal formation between the aorta and pulmonary artery, as well as the patterning of semilunar valve leaflets and the cardiac cushions, the structure which gives rise to the semilunar valves [48,49]. Recently cNCCs have been demonstrated to contribute to the cardiac musculature, where fluorescently labelled cNCCs were stably integrated into the ventricular

myocardium and began expressing Troponin and Myosin Heavy Chain in chick and mouse embryos [50]. Additionally, these labelled cNCCs were seen in the cardiac cushion. This is important, as the semilunar valve remodelling complications presented in our work were not ascribable to myocardial *Shroom3* loss, as the cardiac cushion is endocardial in origin [45,51,52]. Supporting this hypothesis, Durbin *et al.*, (2020) documented *Shroom3* expression in the cardiac neural crest cells at E9.5. cNCC loss contributes to many CHDs including outflow tract septation defects, double outlet right ventricle, VSDs, and valve malformations [49,53–59]. Thus, it is possible that cNCCs expressing *Shroom3* may integrate into the myocardium and the cardiac cushions during embryonic development. Loss of *Shroom3* in this cell population, rather than *Shroom3* arising from cardiac mesodermal cells, may be sufficient to produce the CHDs described here. Supporting this, *Nkx2–5* is not expressed in the cNCCs of the mouse embryo, thus our conditional knockout model of *Shroom3* would not have targeted this specific subset of cells. This evidence allows for the conclusion that *Shroom3* in the heart does not act in a cell autonomous manner, and that external cell types which also express *Shroom3*, particularly the cNCCs, likely contribute to the proper development of the heart. It should be noted that while we were able to show high levels of recombination, we cannot formally rule out that a small number of myocardial cells may still be expressing *Shroom3* and could contribute to cardiac morphogenesis.

### *Shroom3* loss causes left ventricular thinning and decreased cardiomyocyte size

Finally, we assessed the impact of this developmental loss on the adult heart. After full body *Shroom3* loss, three-month and eight-month-old mice showed significant thinning in the left ventricle, a phenotype which continued from development. This was found to be due to decreased cardiomyocyte cross-sectional area compared to littermate controls. However, in adult hearts with myocardial specific loss of *Shroom3* during development, neither left ventricular thinning, nor smaller cardiomyocytes, were seen. Again, this was similar to developmental phenotypes which also found no ventricular thinning. The best-established function of SHROOM3, altering apical-basal cell shape in epithelial cells, is not seen in the myocardium. Rather, myocardial cells are polarized via the planar cell polarity pathway (PCP) [60–62]. Direct functional interactions between SHROOM3 and the PCP pathway during neural tube closure have been identified, as SHROOM3 is directly associated with Lrp2 and Dishevelled2 [14,63]. This interaction between Dishevelled2 has already been validated in the heart [6]. It is then possible that this interaction between SHROOM3 and the PCP pathway may support a function for SHROOM3 in actin organization within the cell, rather than causing cell shape changes in an apical-basal polarity system. This is supported by findings of SHROOM3 function during axon development, where SHROOM3 has been shown to regulate the projection of the axons through the recruitment and organization of F-actin [64]. Thus, actin organization controlled by SHROOM3 may allow for proper heart muscle development and maintenance of shape through the polarization and cell structure pathways.

Paramount to following this work, a Cre recombinase line should be used to selectively knock out *Shroom3* in the cNCCs during development. Markers for this cell type, particularly *Wnt1*, are readily available. It is currently unknown if SHROOM3 is implicated in the induction or migration of the cNCCs. Further investigation is needed to determine the role of SHROOM in the cNCCs, and how this cell population impacts heart development. Additionally future analysis should assess the three-dimensional structure of the heart as initial morphogenesis of the organ can contribute to later maturation and development of the structure [36]. Cardiac functionality, including internal pressure changes, affect later cellular proliferation [46,65]. Thus, to understand the true geometry and arrangements of the cells and their actin networks, and to better understand the consequences of *Shroom3* loss, 3D reconstruction is needed.

## Conclusion

From this data, we conclude that while SHROOM3 is an important contributor to mammalian heart development and postnatal heart structure, SHROOM3 derived from the myocardium of the developing heart is not the sole contributor. While loss of *Shroom3* in the entire developing mouse body resulted in CHDs, including ventricular and septal defects,

semilunar valve defects, and smaller cardiomyocytes, myocardial specific loss of *Shroom3* during development has no impact on heart morphology developmentally or postnatally. In this report, we have also presented the verification that our novel *Shroom3^fl^* line produces a null allele upon recombination, and that this allele can be activated in a temporally and spatially specific manner.

## Supporting information

**S1 Fig. Schematic of *Shroom3* gene trap allele and novel floxed *Shroom3* allele.** A) Schematic of *Shroom3* gene trap line B6.129S4-Shroom3^Gt(ROSA53)Sor/J^. The inserted cassette includes a Splice Acceptor site (SA), an E.coli lacZ, Cre recombinase (Cre), and a polyadenylation sequence (pA). This inserted cassette is under control of the endogenous *Shroom3* promoter. Insertion between exon 3 and exon 4 prevents proper mRNA formation, preventing functional protein from being made. B) Schematic of the novel floxed *Shroom3* allele, created by Cyagen. LoxP sites were inserted to flank exon 5 of the endogenous *Shroom3* gene. Upon recombination, the constitutive knockout allele was designed to produce a null allele.
(TIF)

**S2 Fig. Semi-quantitative PCR for *Shroom3^fl^* recombination and band quantification with densitometry.** A) Primers were designed to detect recombination of the floxed *Shroom3* allele recombination. Forward and reverse primers create 181 bp band (lower band, right gel). Forward and knockout primers create 306 bp band (upper band, right gel). Samples from neonate and 10m adult mouse heart, with kidney for tissue specificity B) cDNA samples from E18.5 hearts. Genotypes for samples are indicated. Samples were run with F-R primer sets and F-Ko primer sets. C) Average band density from F-R primer set, compared to band density from F-Ko. n = 4. F = Forward, R = Reverse, Ko = Knockout, H = Heart, K = Kidney. Raw gel images and lane annotations can be found at Open Science Framework (https://doi.org/10.17605/OSF.IO/NUMF2).
(TIF)

**S3 Fig. Additional images of *Nkx2–5-Cre;Shroom3^fl/fl^* neonate semilunar valve morphologies.** Images of semilunar valves taken from neonate mice from the *Nkx2–5-Cre;Shroom3^+/fl^* X *Shroom3^fl/fl^* cross. Aortic valves are displayed on the left and pulmonary valves are displayed on the right. Images present differing angles and depths of sectioning. Sectioned in the frontal plane.
(TIF)

## Acknowledgments

We thank the lab of Qingping Feng for their donation of the *Nkx2–5* Cre recombinase line.

## Author contributions

**Conceptualization:** Jennifer L. Carleton, Rami R. Halabi, Thomas A. Drysdale.

**Data curation:** Jennifer L. Carleton, Rami R. Halabi, Timothy F. Plageman Jr..

**Formal analysis:** Jennifer L. Carleton, Rami R. Halabi.

**Investigation:** Jennifer L. Carleton.

**Methodology:** Jennifer L. Carleton, Rami R. Halabi, Thomas A. Drysdale.

**Resources:** Timothy F. Plageman Jr., Darren Bridgewater, Qingping Feng.

**Supervision:** Thomas A. Drysdale.

**Validation:** Jennifer L. Carleton, Rami R. Halabi.

**Writing – original draft:** Jennifer L. Carleton, Rami R. Halabi.

**Writing – review & editing:** Jennifer L. Carleton, Jessica A. Willson.

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
