## [Decision Letter · Decision Letter 0]

16 Apr 2025

PONE-D-25-16398Loss of Shroom3 in the developing mouse myocardium does not impact heart developmentPLOS ONE

Dear Dr. Carleton,

Thank you for submitting your manuscript to PLOS ONE. After careful consideration, we feel that it has merit but does not fully meet PLOS ONE’s publication criteria as it currently stands. Therefore, we invite you to submit a revised version of the manuscript that addresses the points raised during the review process.

Specifically, the reviewers have significant concerns about the small dataset used and suggest to refocus the manuscript on the novel findings obtained with a robust dataset using an established model. Further, they also ask to add updated citations on human heart defects in SHROOM3.

We look forward to receiving your revised manuscript.

Kind regards,

Federica Limana

Academic Editor

PLOS ONE

Reviewers' comments:

Reviewer's Responses to Questions

**Comments to the Author**

1. Is the manuscript technically sound, and do the data support the conclusions?

Reviewer #1: Yes

Reviewer #2: Yes

2. Has the statistical analysis been performed appropriately and rigorously?

Reviewer #1: Yes

Reviewer #2: Yes

3. Have the authors made all data underlying the findings in their manuscript fully available?

Reviewer #1: Yes

Reviewer #2: Yes

4. Is the manuscript presented in an intelligible fashion and written in standard English?

Reviewer #1: Yes

Reviewer #2: Yes

5. Review Comments to the Author

Reviewer #1: SHROOM3 expression has been shown in the developing heart where it has been shown to play an important role in heart development, although the cell type responsible for this effect is not known and is the focus of the study.

The authors generated a floxed Shroom3 mouse line to test loss of Shroom3 specifically in myocardial cells. The overall conclusion is that loss of Shroom3 expression in these cells is not responsible for defects in cardiac development associated with full body loss of Shroom3.

Specific findings:

• Shroom expression starting at E10.5 in developing mice.

• Expression can be detected even earlier within myocardial cells but not in endocardial cells.

• Expression of Shroom fades after birth in the heart overall but remains within the myocardium and within the left atrium.

• The authors show a significant decrease in cardiac cross-sectional area in full body Shroom3 loss but this loss is not recapitulated within the mice where Shroom3 is only lost in the myocardial cells. Similar effects are seen when examining Shroom3 associated heart defects and ventricular thinning.

Overall, the manuscript demonstrates the reported findings. I did have a few minor points that should be taken within the context of trying to make the data more accessible to readers with less background in physiology or tissue development.

Minor points:

In the quantification of compact layer thickness in Figure8C and 8F, the effect is clear between wt and heterozygous Shroom3 GT hearts but this reviewer would appreciate seeing the data points used for these graphs so that I can appreciate how representative the data in the rest of the figure is.

Additionally, is there another measurement that is not expected to change as a result of Shroom3 expression that could be quantified as a negative control?

In a related note, for measurements of heart wall thickness, is there a way to normalize these values to overall heart size or dimension so that we can be sure that the effect is not due to overall differences in heart size.

For Fig9C, as you are measuring the cross-sectional area for a large number of cells, a violin-plot would allow us to see the distribution of the data. Once again, I have no doubt as to the effect being shown, but I’d love to see if the effect was uniform for all cells or just a subset of the population (presumably a large subset of the population). This is more of a concern in Figure15 however.

For figure13, I appreciate that the point is to measure the thickness of both left and right ventricular walls in either wt of myocardial shroom loss mice but the placement of the lines used for measurements does not appear to reflect the actual thickness of the wall in 13B. The difference is dramatic and would change the conclusions for the figures. Since this is not my specific area of expertise, this reviewer would appreciate more explanation for how the boundaries are chosen (especially for the inside boundary). Once again, data points on the bar graphs would also help a reader appreciate the data with more nuance.

Reviewer #2: The manuscript by Carleton et al describe an important role for a novel candidate in human disease called SHROOM3. SHROOM3 has been implicated in human disease including primarily kidney defects, and more recently, cardiac defects. A manuscript from 2020 describes heart defects due to SHROOM3 loss of function in embryos. Carleton et al. have expanded these findings, describing a more detailed expression pattern, additional cardiac defects, including valve defects, and pathology in adult cardiomyocytes, with smaller cardiomyocytes persisting to adulthood. These findings are important to our understanding of cardiac development and relevant to human disease. In terms of placing this novel data in the literature, the manuscript describes a role for SHROOM3 in patients with heterotaxy, however a brief literature review reveals two more recent studies, that have shown a role for SHROOM3 in human cardiac disease, including patients with CHD (PMID: 36011280, PMID: 39202774) and upregulation of SHROOM3 after myocardial infarction (PMID: 39086770). The findings of Carleton et al. may have direct relevance to these studies, given their findings of left ventricle thinning and valve defects. Further their findings in adult mice, in terms of the expression pattern and smaller cardiomyocyte area, are relevant to upregulation of SHROOM3 post myocardial infarction. The study needs to highlight these more recent findings and describe their data in the full context of available literature surrounding SHROOM3. Carleton et al. also describe, for the first time, a SHROOM3 conditional loss of function mouse-line. However, this dataset is much smaller. Carleton et al describe that loss of function in the myocardium does not lead to heart defects. These findings may be important for mechanism, and the authors highlight that other cell lines like neural crest are likely contributors to cardiac defects. This finding is important data in narrowing SHROOM3‘s lineage-specific role. However, the conclusions are based on the evaluation of a small number of embryos. This small number is likely not sufficient given the partial penetrance of the described heart defects, and especially to draw primary conclusions for the manuscript. Carleton et al need to better highlight what has been previously reported in SHROOM3 loss of function mice, and the novel findings added here (which are important.) The manuscript would be stronger with enlarging this small dataset or refocusing onto their important findings in a more robust dataset using an established model. Below are comments which, if addressed, would make the manuscript suitable for publication.

1. SHROOM3 loss of function has been shown to result in heart defects in mouse embryos, however the current study adds novel findings. The study needs to describe what has been previously reported and what are the novel findings. The novel findings need to be highlighted. This includes: 1) Evaluation of embryos at e18.5, (versus 14.5 in the Durbin et all study, where E18.5 represent a more robust timepoint for evaluation of cardiac defects than Durbin et al.) 2) A more detailed expression pattern is described 3) A spectrum of valve defects are reported – for the first time. 4) Ventricle thinning is isolated to left ventricle 5) ventricle thinning persists to adulthood. 6) They measure cardiomyocyte area and the smaller size persists into adulthood

2. The manuscript needs to cite more recent literature PMID: 36011280, PMID: 39202774, PMID: 39086770), and place their data in context (for example, their findings related to SHROOM3 valve defects and ventricle thinning may be relevant in hypoplastic left heart syndrome patients, and SHROOM3 expression pattern post-myocardial infarction.

3. The manuscript presents a robust dataset from an established gene trap model. They then present a smaller dataset using a conditional loss of function approach. The later findings are important, given they are the first lineage-specific deletion of SHROOM3 reported. However, the dataset is much smaller than the evaluation of nearly 100 gene trap embryos. From my reading, the conclusions are based on n=8 flox/flox embryos for cardiac defects and n=3 for wall thickness. The results section needs to make more transparent how many embryos were analyzed in each group. Furthermore, 8 embryos does not seam sufficient to make a conclusion, given the partial penetrance of the cardiac defects. These numbers need to be increased, or, at least, the data needs to be described objectively without such strong conclusion.

4. The authors state their findings indicate other cell populations, like the neural crest cells, may be important for heart defects. Are neural crest cells important to cardiomyocyte size and ventricle thickness? Perhaps myocardial SHROOM3 expression is important in conjunction with other cell populations like the cardiac neural crest? If the authors have data about conditional loss of function using a Wnt1-Cre resulting in the neural crest deletion of SHROOM3, they should report it here.

5. To the best of their ability, the authors should refocus the manuscript on the novel findings obtained with a robust dataset using an established model, versus findings from a smaller number of embryos using a new model.

6. The title of the article needs to be modified to highlight the novel findings in the manuscript, obtained from a robust dataset, versus the conclusions drawn from a small cohort of embryos. Perhaps: (as stated in the first line of the conclusion), “SHROOM3 is an important contributor to mammalian heart development and postnatal heart structure.” or “The role of SHROOM3 in cardiac defects.”

7. The abstract also needs to be modified to highlight the novel findings obtained from a robust dataset versus the negative conclusions drawn from a much smaller number of embryos.

The manuscript presents important and novel findings. However, a better description of the numbers utilized in the dataset is needed. Furthermore, if possible, the numbers for the conditional loss of function embryos should be increased, and a second lineage-specific deletion should be performed. At the very least, they should describe their novel findings in the context of what was previously demonstrated in SHROOM3 loss of function embryo heart defects. The authors should also add updated citations on human heart defects in SHROOM3, properly placing the data in the literature. Overall, the authors should refocus the manuscript on the novel findings obtained with a robust dataset using an established model. These findings may have clinical relevance in heart disease given recent findings in patients.

6. PLOS authors have the option to publish the peer review history of their article (what does this mean? ). If published, this will include your full peer review and any attached files.

**Do you want your identity to be public for this peer review?** For information about this choice, including consent withdrawal, please see our Privacy Policy .

Reviewer #1: No

Reviewer #2: No

---

## [Author Response · Author response to Decision Letter 1]

14 Jul 2025

Dear Dr. Chenette:

We would like to thank the reviewers for their constructive comments and suggestions. Their feedback is much appreciated. We have responded to the comments from Reviewer #1 and amended certain figures in our manuscript. The discussion and citations have been updated as requested by Reviewer #2, and n-values have been increased. Specific responses and updates are as follows.

Reviewer #1

1. In the quantification of compact layer thickness in Figure8C and 8F, the effect is clear between wt and heterozygous Shroom3 GT hearts but this reviewer would appreciate seeing the data points used for these graphs so that I can appreciate how representative the data in the rest of the figure is.

We agree that inclusion of the data points on all graphs in the manuscript is beneficial for supporting our conclusions and better appreciating the data. Unfortunately, this data was collected several years ago, and we have been unable to locate our original individual measurement values from the initial Shroom3Gt analysis. The student who did these measurements has graduated from the lab, and our computer for original data storage was corrupted several years ago. We have not been able to find those values on other lab computers. While this is clearly an issue, we can provide further corroborating data in the original thesis on this topic (https://ir.lib.uwo.ca/etd/1394/). Included in the thesis are additional ventricular wall measurements at E16.5, E17.5, and wall thickness measurements using in vitro ultrasound which all show significant thinning in the left ventricle in Shroom3Gt/Gt mice. In addition, analysis of the same mice in Durbin et al., 2020 also observe left ventricular wall thinning at E14.5.

In this manuscript, the most important point is that the myocardial specific knockout does not produce ventricular thinning. While we can provide the data points for that observation, we have not done so to maintain consistency with the Shroom3Gt/Gt figures. If reviewers feel otherwise, we would be willing to provide the scatter plot for that data.

2. Additionally, is there another measurement that is not expected to change as a result of Shroom3 expression that could be quantified as a negative control? In a related note, for measurements of heart wall thickness, is there a way to normalize these values to overall heart size or dimension so that we can be sure that the effect is not due to overall differences in heart size.

While this would be interesting, normalization of heart measurements is not normally done in the field of heart development and in our experience, it is not straightforward. When our embryo and neonate hearts are collected, we keep the torso (everything inside the ribcage) intact. While this has allowed us to assess any looping or symmetry defects, it precludes us from taking heart weight measurements. When these torsos are embedded in paraffin wax, although we do our best to embed in the same orientation, small changes to the angle of embedding do change the final viewing of the organ, even when sectioned in the same plane. One parameter which could be used for normalization is septal wall thickness. However, this measurement is subject to change depending on the angle of embedding, making it unreliable and comparisons difficult to interpret.

We have done measurements of total heart weight in adult mice and have not observed any effect based on genotype (Fig. 14).

3. For Fig9C, as you are measuring the cross-sectional area for a large number of cells, a violin-plot would allow us to see the distribution of the data. Once again, I have no doubt as to the effect being shown, but I’d love to see if the effect was uniform for all cells or just a subset of the population (presumably a large subset of the population). This is more of a concern in Figure15 however.

We do agree that a violin plot would better represent the data for this type of measurement. However, similarly to the response in comment 1, we have been unable to recover the original individual data points for the Shroom3Gt experiments. Making a violin plot would be possible for data from the Shroom3fl experiments, but again, for the sake of consistency we have chosen to present the Shroom3fl data in the same style as the Shroom3Gt data.

4. For figure13, I appreciate that the point is to measure the thickness of both left and right ventricular walls in either wt of myocardial shroom loss mice but the placement of the lines used for measurements does not appear to reflect the actual thickness of the wall in 13B. The difference is dramatic and would change the conclusions for the figures. Since this is not my specific area of expertise, this reviewer would appreciate more explanation for how the boundaries are chosen (especially for the inside boundary). Once again, data points on the bar graphs would also help a reader appreciate the data with more nuance.

We appreciate the reviewer’s comments as this is a common issue in the field and defining the exact boundary is often difficult. In this study, we have measured the thickness of the ventricular wall at the midpoint along the length of the ventricle. This is consistent with other studies in the field. Ventricular thickness can vary at different parts of the heart, particularly near the apex, and so the midpoint is often used for consistency. The compact myocardium is the contractile musculature where individual cells are aligned with the length of the heart. This area is densely packed, where the trabecular layer consists of finger-like, muscular projections that extend into the chamber of the heart. They are usually aligned more perpendicular to the length of the heart with gaps between the projections.

The example image used in Fig. 13 also had portions of the papillary muscle visible. These are muscles inside the ventricles which anchor the leaflets of the valves. We have used a new, hopefully clearer, example image in our revision. The image has been updated, and the delineation of the compact myocardium from the trabeculae has been indicated with a dotted line. For consistency, example images in Fig. 6 and Fig. 8 have been updated in the same way.

Reviewer #2

1. SHROOM3 loss of function has been shown to result in heart defects in mouse embryos, however the current study adds novel findings. The study needs to describe what has been previously reported and what are the novel findings. The novel findings need to be highlighted. This includes: 1) Evaluation of embryos at e18.5, (versus 14.5 in the Durbin et all study, where E18.5 represent a more robust timepoint for evaluation of cardiac defects than Durbin et al.) 2) A more detailed expression pattern is described 3) A spectrum of valve defects are reported – for the first time. 4) Ventricle thinning is isolated to left ventricle 5) ventricle thinning persists to adulthood. 6) They measure cardiomyocyte area and the smaller size persists into adulthood.

We thank the reviewer for these robust suggestions. A more in-depth comparison between our findings and those presented in Durbin et al., (2020) has been written into the discussion. These can be found on page 21, line 501-503, and page 21-22, line 512-520.

2. The manuscript needs to cite more recent literature PMID: 36011280, PMID: 39202774, PMID: 39086770), and place their data in context (for example, their findings related to SHROOM3 valve defects and ventricle thinning may be relevant in hypoplastic left heart syndrome patients, and SHROOM3 expression pattern post-myocardial infarction.

Again, we thank the reviewer for taking the time to provide resources for us. These citations have been included on page 3, line 47 for PMID: 36011280, and page 20, line 483-486 for PMID: 39086770.

After reading through PMID: 39202774 and their citations, we have chosen not to include this in our manuscript. They rely on citations from Durbin et al., (2020), which we reference and discuss in greater detail.

3. The manuscript presents a robust dataset from an established gene trap model. They then present a smaller dataset using a conditional loss of function approach. The later findings are important, given they are the first lineage-specific deletion of SHROOM3 reported. However, the dataset is much smaller than the evaluation of nearly 100 gene trap embryos. From my reading, the conclusions are based on n=8 flox/flox embryos for cardiac defects and n=3 for wall thickness. The results section needs to make more transparent how many embryos were analyzed in each group. Furthermore, 8 embryos does not seam sufficient to make a conclusion, given the partial penetrance of the cardiac defects. These numbers need to be increased, or, at least, the data needs to be described objectively without such strong conclusion.

We appreciate the recognition that this is the first lineage-specific knockout of Shroom3 in the heart. Please note that a retina-specific knockout of Shroom3 done by a collaborator using the same floxed line has recently been published (Herstine et al., 2025).

In terms of the n-vale discrepancy between Shroom3Gt and Shroom3fl experiments, we concede that there certainly is a significant difference. In the Shroom3Gt experiments, a maximum of 24 Shroom3+/+, 24 Shroom3+/Gt, and 33 Shroom3Gt/Gt E18.5 embryos were assessed for morphological defects. This contrasts with 8 control, 8 Nkx2-5-Cre;Shroom3+/fl, and 8 Nkx2-5-Cre;Shroom3fl/fl neonates that were assessed in Shroom3fl experiments. We agree that this was a concern and so we have added 30 animals to our Shroom3fl study, increasing the final n-values for the Shroom3fl experiment to 23, 8, and 23 neonates, respectively per the genotypes above. We currently maintain our Shroom3fl mouse line in a homozygous state, and breeding the heterozygous allele back into our colony would have taken too much time. In addition, the lack of phenotype in the homozygous floxed mice makes a phenotype in the heterozygous floxed mice very unlikely. The increased n-values for CHDs in the floxed line can be found in the methods section on page 7, line 157 and page 8, line 164-165. We still did not observe any CHDs in the floxed mice and have added more representative images in supplemental figure 3. Additionally, n-values for Shroom3fl ventricle wall thickness have been increased to 6, which is now the same as Shroom3Gt ventricle wall thickness at E18.5. This can be found in the methods section on page 8, line 178-179. The bar chart in Fig. 13 has been updated accordingly.

In the results section, we have explicitly written n-values in each experiment for morphological assessments. These changes can be found on page 12, line 270, 278, and 279. This will hopefully increase the transparency of the numbers assessed in these experiments. As well, for all morphology assessments, n-vales for each genotype are written in the figure legends.

We would also like to present a further description of our mice to support the lack of phenotypes in the Shroom3fl line. Nearly all the mice in our NkxCre;Shroom3fl colony live until a year of age. With the caveat that there may still be very low levels of functional protein in the heart based on the levels of recombination we see, the high levels of survival suggest no physiological implications from the lack of Shroom3.

4. The authors state their findings indicate other cell populations, like the neural crest cells, may be important for heart defects. Are neural crest cells important to cardiomyocyte size and ventricle thickness? Perhaps myocardial SHROOM3 expression is important in conjunction with other cell populations like the cardiac neural crest? If the authors have data about conditional loss of function using a Wnt1-Cre resulting in the neural crest deletion of SHROOM3, they should report it here.

The neural crest is the most obvious candidate for a cell lineage that could have roles in cardiac morphogenesis and wall thickness and still have Shroom3 expression based on the expression of the Nkx2-5 Cre driver. There has been evidence that cardiac neural crest cells integrate into the musculature of the heart and take on a myocardial identity. We highlight this in the discussion. However, from the available literature, there has been no evidence to suggest that neural crest cells are important in the size of cardiomyocytes, nor do they contribute to the thickening of the ventricle.

In the conditional knockout model we have created, we have eliminated the majority of myocardial Shroom3 expression and have found no phenotypes arising from this loss. That is why we suggest other cell populations with Shroom3 are responsible for the defects seen in the Shroom3Gt. We do agree that the cells which integrate into the musculature of the heart must be working in conjunction with other cell types which arise from the heart itself. While we do not have data ready to present in this manuscript, we are in the process of working on collaborations to follow these results with a Wnt1 Cre recombinase line.

5. To the best of their ability, the authors should refocus the manuscript on the novel findings obtained with a robust dataset using an established model, versus findings from a smaller number of embryos using a new model.

While we understand the reasoning behind this suggestion, we hope that the amendments outlined in the response to comment 3 will give support to equally focus on both the established model and the new model. Since the time that we submitted this manuscript, our collaborators have published their findings of eye development using the same Shroom3fl line. This citation has been included on page 5, line 99-100 and page 15, line 354 and 363. They have found that Shroom3 is implicated in optic fissure closure and in regulating cellular polarity in these tissues. We believe that this adds credibility and validity of the new mouse model.

In addition, we strongly believe that this new model is essential to the story and intent of the manuscript. The conclusions we have made add to an emerging understanding of the cardiac neural crest cells and their role in heart development. They also provide a poignant example of non-cell autonomous influences on organogenesis and morphogenesis. This is valuable to the field of developmental biology as a whole.

6. The title of the article needs to be modified to highlight the novel findings in the manuscript, obtained from a robust dataset, versus the conclusions drawn from a small cohort of embryos. Perhaps: (as stated in the first line of the conclusion), “SHROOM3 is an important contributor to mammalian heart development and postnatal heart structure.” or “The role of SHROOM3 in cardiac defects.”

Similar to the above comment, we strongly believe that both data sets together tell a more complete and novel story. We hope that the amendments to the n-values in the Shroom3fl line lend credence to this. A title that focusses on the Shroom3Gt data set may be too similar to Durbin et al., (2020). As an alternative, we have opted for a more generic title: Understanding the role of Shroom3 in the developing mouse myocardium

7. The abstract also needs to be modified to highlight the novel findings obtained from a robust dataset versus the negative conclusions drawn from a much smaller number of embryos.

In the same vein as both above responses, we believe that both mouse lines highlight important findings which should be presented. Once again, we are hopeful that the additional n-values will support our decision.

Again, we thank the reviewers for their very timely responses and thorough feedback.

Sincerely,

Jennifer Carleton

PhD candidate, Drysdale lab

Department of Physiology and Pharmacology, Developmental Biology specialization

Schulich School of Medicine and Dentistry

The University of Western Ontario

---

## [Decision Letter · Decision Letter 1]

31 Jul 2025

PONE-D-25-16398R1Understanding the role of Shroom3 in the developing mouse myocardiumPLOS ONE

Dear Dr. Carleton,

Thank you for submitting your manuscript to PLOS ONE. After careful consideration, we feel that it has merit but does not fully meet PLOS ONE’s publication criteria as it currently stands. Therefore, we invite you to submit a revised version of the manuscript that addresses the points raised during the review process.

There are still concerns about the presence of Shroom 3 in the outflow tract 

We look forward to receiving your revised manuscript.

Kind regards,

Federica Limana

Academic Editor

PLOS ONE

Journal Requirements:

Reviewers' comments:

Reviewer's Responses to Questions

**Comments to the Author**

1. If the authors have adequately addressed your comments raised in a previous round of review and you feel that this manuscript is now acceptable for publication, you may indicate that here to bypass the “Comments to the Author” section, enter your conflict of interest statement in the “Confidential to Editor” section, and submit your "Accept" recommendation.

Reviewer #1: All comments have been addressed

Reviewer #2: (No Response)

2. Is the manuscript technically sound, and do the data support the conclusions?

Reviewer #1: (No Response)

Reviewer #2: Yes

3. Has the statistical analysis been performed appropriately and rigorously?

Reviewer #1: (No Response)

Reviewer #2: Yes

4. Have the authors made all data underlying the findings in their manuscript fully available?

Reviewer #1: (No Response)

Reviewer #2: Yes

5. Is the manuscript presented in an intelligible fashion and written in standard English?

Reviewer #1: (No Response)

Reviewer #2: Yes

6. Review Comments to the Author

Reviewer #1: (No Response)

Reviewer #2: The increased number of embryos analyzed, clarification of the number of embryos used in each analysis, and the revised title, abstract and discussion satisfy previous concerns.

One minor point that needs to be addressed before acceptance.

The manuscript describes the Shroom3 expression pattern, which is robust and an important contribution to the manuscript. The manuscript states: “This expression was specific to the myocardium of the atria and ventricles of the heart and was not seen in the outflow tracts or great arteries….” and then later …” Shroom3 expression was seen in the base of the outflow tract“

The staining pattern is clear and there is clearly a sharp drop-off and absence in the great arteries. However, it seems contradictory to say staining is not present in the outflow tract and then later that it is present in the base of the outflow tract. It seems more accurate to state that staining was localized to the base of the outflow tract and absent from the great arteries.

This is an important point given that cardiac neural crest cells are responsible for outflow tract, aortopulmonary, and ventricular septation, and the article hypothesizes (logically) that neural crest cells may be responsible for the septal defects seen in whole body loss of function embryos.

Clarification of this minor point would make the manuscript acceptable for publication.

7. PLOS authors have the option to publish the peer review history of their article (what does this mean? ). If published, this will include your full peer review and any attached files.

**Do you want your identity to be public for this peer review?** For information about this choice, including consent withdrawal, please see our Privacy Policy .

Reviewer #1: No

Reviewer #2: No

---

## [Author Response · Author response to Decision Letter 2]

11 Aug 2025

Dear Dr. Chenette:

We are happy to respond to Reviewer #2 and address their concerns about the description of Shroom3 expression in the outflow tract.

Reviewer #2

1. The manuscript describes the Shroom3 expression pattern, which is robust and an important contribution to the manuscript. The manuscript states: “This expression was specific to the myocardium of the atria and ventricles of the heart and was not seen in the outflow tracts or great arteries….” and then later …” Shroom3 expression was seen in the base of the outflow tract“

The staining pattern is clear and there is clearly a sharp drop-off and absence in the great arteries. However, it seems contradictory to say staining is not present in the outflow tract and then later that it is present in the base of the outflow tract. It seems more accurate to state that staining was localized to the base of the outflow tract and absent from the great arteries.

This is an important point given that cardiac neural crest cells are responsible for outflow tract, aortopulmonary, and ventricular septation, and the article hypothesizes (logically) that neural crest cells may be responsible for the septal defects seen in whole body loss of function embryos.

Clarification of this minor point would make the manuscript acceptable for publication

In saying “the base of the outflow tracts”, we were referring to the portion of the heart which surrounds the base of the outflow tracts, rather than the most basal portion of the outflow tract itself. We understand that our description was unclear and we thank you for bringing it to our attention. To make this clearer, we have amended the sentences to read “the myocardium surrounding the base of the outflow tracts”. These changes can be found on page 10, line 219-220; page 11, line 237-238, 254; and page 20, line 475.

We once again thank the reviewers for their attention to detail and their suggestions and insights which have improved the quality of the manuscript.

Sincerely,

Jennifer Carleton

PhD candidate, Drysdale lab

Department of Physiology and Pharmacology, Developmental Biology specialization

Schulich School of Medicine and Dentistry

The University of Western Ontario

---

## [Editor Report · Decision Letter 2]

19 Aug 2025

Understanding the role of Shroom3 in the developing mouse myocardium

PONE-D-25-16398R2

Dear Dr. Carleton,

We’re pleased to inform you that your manuscript has been judged scientifically suitable for publication and will be formally accepted for publication once it meets all outstanding technical requirements.

Kind regards,

Federica Limana

Academic Editor

PLOS ONE
---

## [Editor Report · Acceptance letter]

PONE-D-25-16398R2

PLOS ONE

Dear Dr. Carleton,

I'm pleased to inform you that your manuscript has been deemed suitable for publication in PLOS ONE. Congratulations! Your manuscript is now being handed over to our production team.

Kind regards,

on behalf of

Dr. Federica Limana

Academic Editor

PLOS ONE